# The behavioral signature of stepwise learning strategy in male rats and its neural correlate in the basal forebrain

Hachi E. Manzur[1], Ksenia Vlasov[1], You-Jhe Jhong [2], Hung-Yen Chen[2] & Shih-Chieh Lin [1,2,3] ✉

Studies of associative learning have commonly focused on how rewarding outcomes are predicted by either sensory stimuli or animals' actions. However, in many learning scenarios, reward delivery requires the occurrence of both sensory stimuli and animals' actions in a specific order, in the form of behavioral sequences. How such behavioral sequences are learned is much less understood. Here we provide behavioral and neurophysiological evidence to show that behavioral sequences are learned using a stepwise strategy. In male rats learning a new association, learning started from the behavioral event closest to the reward and sequentially incorporated earlier events. This led to the sequential refinement of reward-seeking behaviors, which was characterized by the stepwise elimination of ineffective and non-rewarded behavioral sequences. At the neuronal level, this stepwise learning process was mirrored by the sequential emergence of basal forebrain neuronal responses toward each event, which quantitatively conveyed a reward prediction error signal and promoted reward-seeking behaviors. Together, these behavioral and neural signatures revealed how behavioral sequences were learned in discrete steps and when each learning step took place.

Associative learning is essential for survival and allows animals and humans to predict future reward based on environmental stimuli[1,2] or their own actions[3–5]. Understanding the algorithmic principles of associative learning has been a central question in psychology and neuroscience[6–13], and has broad implications in machine learning and artificial intelligence[14–16].

While the learning of stimulus-reward and action-reward associations have been historically studied under the separate labels of Pavlovian[1,2] and instrumental[3,4] conditioning, most learning scenarios require the synergistic contribution from both types of learning strategies. For example, when a new reward-predicting stimulus is introduced to the environment, the Pavlovian strategy might not be sufficient because oftentimes the reward would not be delivered unless animals take specific actions. In experimental

settings, such actions could be a lever press, or a saccade toward a target, or multiple licks before the reward is delivered. In these scenarios, reward is obtained only when sensory stimuli and animals' actions occur in a specific order as a behavioral sequence[5,17–19]. Compared to the wealth of knowledge about Pavlovian and instrumental conditioning, little is understood about how animals learn behavioral sequences that contain both stimuli and actions.

Converging views from theoretical studies support the idea that reward-predicting behavioral sequences can be efficiently learned using a strategy that we will refer to as stepwise learning: learning starts from the event closest to the reward, while earlier events are learned in later steps. This learning strategy was initially proposed by Skinner[5] and more recently elaborated into formal learning models[18,19].

[1]Neural Circuits and Cognition Unit, Laboratory of Behavioral Neuroscience, National Institute on Aging, National Institutes of Health, Baltimore, MD, USA. [2]Institute of Neuroscience, National Yang Ming Chiao Tung University, Taipei, Taiwan. [3]Brain Research Center, National Yang Ming Chiao Tung University, Taipei, Taiwan. ✉e-mail: shihchieh.lin@nycu.edu.tw

Similar learning dynamics are also predicted by reinforcement learning algorithms, in which states that are closer to the final reward are learned first[14]. The stepwise learning strategy has also been successfully used in various animal training scenarios to incrementally chain single behaviors into long sequences over multiple training steps[20].

The goal of the current study is to test whether animals use the stepwise learning strategy to learn reward-predicting behavioral sequences that contain both stimuli and actions. We seek to identify the behavioral and neural signatures of this learning process that can delineate the discrete steps of learning. A major challenge in understanding this type of learning is that behavioral sequences are controlled not only by the experimenter but also by the animal, which is free to take various actions. We reason that, at the beginning of learning, animals' actions would be less constrained and therefore would generate a large repertoire of behavioral sequences that may or may not lead to the rewarding outcome. As the stepwise learning process unfolds, the repertoire of behavioral sequences should become increasingly selective as well as more frequently rewarded. Therefore, the behavioral signature of the stepwise learning strategy may reside in how the entire repertoire of behavioral sequences become sequentially refined during the learning process. In the current study, we identified such a behavioral signature, which corresponded to the discrete steps in the stepwise learning process.

In order to validate the behavioral signature for the stepwise learning strategy, we focused on a special subset of noncholinergic neurons in the basal forebrain (BF), which are referred to as BF bursting neurons[21–26]. Previous studies have found that BF bursting neurons convey a reward-prediction error signal[21,22,27,28], and show highly robust phasic bursting responses to reward-predicting sensory stimuli irrespective of their sensory modalities[21–25]. Moreover, such responses only emerge after reward-based associative learning[21], and are tightly coupled with behavioral performance and promotes faster decision speeds[21–23]. These observations suggest that increased BF bursting neuron activities toward a behavioral event reflects that the event has been learned as a reward predictor. By observing the temporal evolution of BF bursting neuron activities throughout the learning process in the current study, we predict that BF responses should mirror the stepwise learning process: BF activity should first emerge toward the last behavioral event closest to the reward, and subsequently develop toward the earlier behavioral events. Such behavioral and neurophysiological findings will provide important insights on how behavioral sequences are learned.

## Results

### A model for learning behavioral sequences using the stepwise learning strategy

To gain intuition about the stepwise learning strategy, we first considered a toy example in which a three-element sequence A-B-C predicted reward (Fig. 1a). This sequence can be learned using the stepwise strategy in three discrete steps, starting from the event closest to the reward and sequentially incorporating earlier events (Fig. 1b). As more behavioral events are learned as reward predictors in each step, only behavioral sequences that contain all the learned events would predict reward and therefore preferentially executed, while incompatible sequences that do not contain all the learned events would not predict reward and therefore be eliminated from the behavioral repertoire. As a result, the discrete steps of learning would correspond to the stepwise elimination of non-rewarded behavioral sequences that share subsets of behavioral elements (Fig. 1b). We hypothesized that this sequential refinement of reward-seeking behaviors might provide a behavioral signature of the stepwise learning strategy.

### Sequential refinement of reward-seeking behaviors during new learning

To test whether animals indeed use the stepwise strategy to learn behavioral sequences that contain both sensory stimuli and their own actions, we trained adult Long-Evans rats in an auditory discrimination task. Rats entered the fixation port to initiate each trial, where they encountered three trial types (S$^{left}$; S$^{right}$; catch) with equal probabilities that respectively indicated sucrose water reward in the left or right port, or no reward in the case of catch trials (no stimulus) (Fig. 2a). During the initial auditory discrimination phase, S$^{left}$ and S$^{right}$ were two distinct sound stimuli. After reaching asymptotic performance, rats entered the new learning phase (first new learning session denoted as the $D_0$ session), in which the S$^{right}$ sound stimulus was switched to a novel light stimulus that minimized sensory generalization from past experience (Fig. 2b and Table 1). During the new learning phase, rats maintained stable levels of performance toward the previously learned S$^{left}$ sound stimulus (Fig. 2c) (94.3 ± 4.8% correct, 109 ± 20 trials per session, mean ± std). Within the first three sessions of the new learning phase, all rats ($N = 7$) began responding correctly in the new light trials (the first such session denoted as the $D_1$ session) and maintained stable levels of >90% correct response rates afterwards (Fig. 2c and Table 2).

To understand how the repertoire of behavioral sequences evolved during learning, we examined all possible behavioral sequences that the animal might experience (Fig. 2d). This approach allowed us to identify non-rewarded behavioral sequences that were

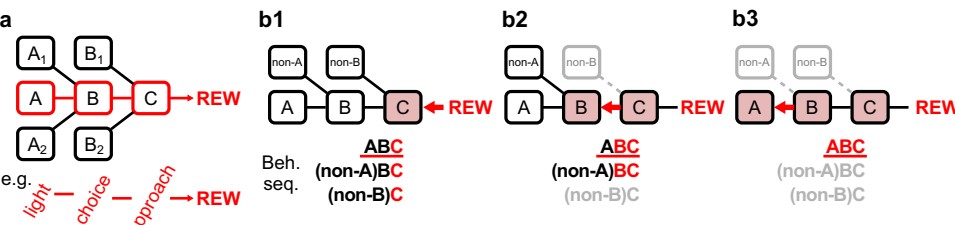

**Fig. 1 | A model for the stepwise learning of behavioral sequences. a** Schematic of an example scenario where the reward is predicted by a simple sequence consisting of three behavioral events A-B-C. For example, A could be a light stimulus, B the rightward choice, and C the approach behavior to obtain reward. A1/A2/B1/B2 indicate alternative behavioral elements for A/B that could be combined to generate other behavioral sequences. Among all possible sequences, only the A-B-C sequence is rewarded. **b** The three distinct steps when learning the A-B-C sequence using the stepwise strategy illustrate how reward-seeking behaviors are sequentially refined. For simplicity, alternative behavioral events are denoted as non-A and non-B. Three possible behavioral sequences are listed. Behavioral events that have

been learned as reward-predictors are colored in red, while the rewarded sequence (A-B-C) is underlined in red. **b1** The first step of learning involves the event closest to the reward, C. Animals would engage in all three behavioral sequences because they all contain this reward-predicting event. **b2** The second step of learning involves the next-to-last event, B. Behavioral sequences that contain the reward-predicting events B-C are preserved, while the incompatible sequence is eliminated from the behavioral repertoire (gray). **b3** The third step of learning involves the earliest event, A. Only the A-B-C sequence contains all the reward-predicting events A-B-C and therefore preserved, while another incompatible sequence is eliminated from the behavioral repertoire.

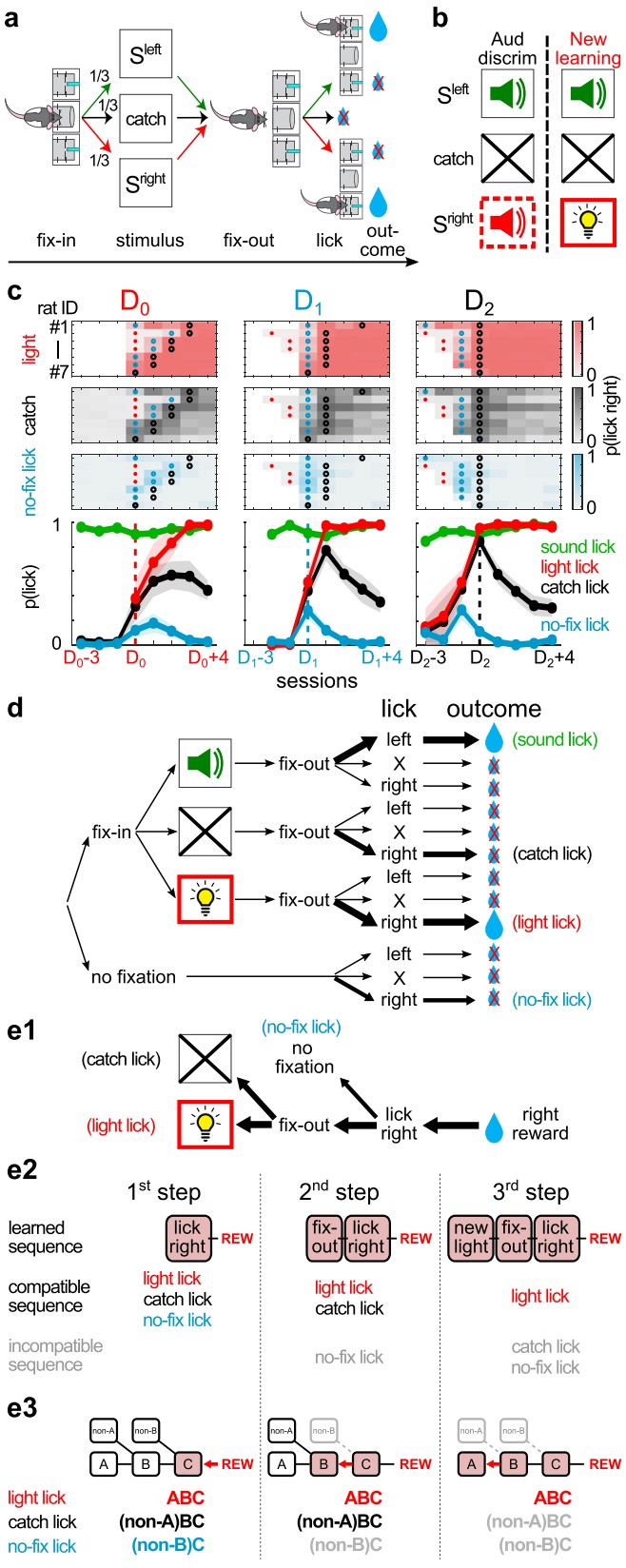

**Fig. 2 | New associative learning led to sequential refinements of reward-seeking behaviors. a** Behavioral task. Rats entered the fixation port to initiate each trial, where they encountered three trial types (S[left]; S[right]; catch) with equal probabilities that respectively indicated water reward in the left or right port, or no reward in the case of catch trials (no stimulus). Task symbols were adapted from Avila and Lin[22]. **b** Rats initially learned an auditory discrimination task (old association phase). At the new learning phase, S[right] was switched from the sound to a new house light, while other elements of the task remained the same. Symbols were adapted from Avila and Lin[22]. **c** The proportion of three types of reward-seeking behaviors toward the right reward port (light licks, catch licks, no-fixation licks) across sessions during new learning in individual animals (*N* = 7 rats), and their mean (±s.e.m.). No-fixation licks (cyan) refers to trials in which rats failed to first enter the fixation port before licking in the right reward port. Sessions were respectively aligned, in each column, at the $D_0$, $D_1$ or $D_2$ session of each animal. $D_0$ refers to the first new learning session with the new light stimulus; $D_1$ refers to the session when animals began to respond correctly in the new light trial; $D_2$ refers to the session when catch licks peaked. The $D_0$, $D_1$ and $D_2$ sessions in each animal were indicated by red, cyan, and black circles, respectively. Each row in top panels depicts behavior performance in one animal (#1–7). In the middle and right panels, only sessions in the new learning phase (starting from the $D_0$ session) were plotted. The emergence, as well as the following sequential elimination, of no-fixation licks and catch licks resembled the sequential refinement of reward-seeking behaviors under the stepwise learning strategy. Source data for this and all subsequent figures are provided as a Source Data file. **d** Behavioral sequences that animals might experience. Out of all possible sequences, only three types of rightward licking behaviors were consistently observed during new learning. Symbols were adapted from Avila and Lin[22]. **e** A stepwise learning model that accounts for the sequential refinement of the three types of rightward licking behaviors. **e1** The behavioral events in the model arranged in the format as in Fig. 1b. **e2** Behavioral sequences learned as reward predictors at the three discrete steps of learning, along with the compatible and incompatible behavioral sequences at each step. **e3** Sequential refinement of the three types of rightward licking behaviors arranged in the format as in Fig. 1b.

and no-fixation licks (Fig. 2c). Catch licks refers to licking responses toward the right reward port in catch trials when no sensory stimulus was presented. No-fixation licks refers to the situation in which rats directly licked at the right reward port without first entering the center fixation port. All three behavioral sequences shared the common feature of licking the right reward port (lick-right).

Both no-fixation licks and catch licks were largely absent before the new learning phase, and emerged and peaked during early sessions of new learning, before subsequently diminished in later sessions (Fig. 2c). No-fixation licks occurred most frequently at the $D_1$ session, while catch licks peaked in the same session (*N* = 1/7) or later in most animals (*N* = 6/7). We will denote the peak of catch licks as the $D_2$ session in each animal. The consistent temporal order of the $D_1$ and $D_2$ sessions within each animal allowed us to identify similar learning stages across animals despite their individual differences in learning dynamics.

The temporal dynamics of the three types of rightward licks during new learning (Fig. 2c) showed that non-rewarded behaviors were sequentially eliminated while rewarded behaviors were preserved. This temporal dynamics resembled the pattern of sequential refinement of behavioral sequences predicted by the stepwise learning strategy (Fig. 2e), and likely corresponded to the discrete steps in the underlying learning process. To further test whether such patterns of sequential refinements represent a general feature of behavioral sequence learning, we trained a separate cohort of animals and observed similar behavioral signatures regardless of the sensory modality of the stimulus or the laterality of the new learning side (Fig. S1). These observations support the idea that animals do use the stepwise strategy to learn behavioral sequences. In the following analyses, we tested additional predictions of the stepwise learning strategy at both behavioral and neurophysiological levels.

not associated with specific trial types but were, nonetheless, highly relevant for the learning process. We identified three types of behavioral sequences whose frequencies consistently increased during the new learning phase. These three types of behavioral sequences included the rewarded licking behavior in the new light trials (light licks), as well as two types of non-rewarded behavioral sequences: catch licks

**Table 1 | Stimulus parameters of the $S^{left}$ and $S^{right}$ stimuli for each animal**

| Animal ID | $S^{left}$ sound | $S^{right}$ sound (auditory discrimination) | $S^{right}$ light (new learning) |
|---|---|---|---|
| #1 | 12 kHz 80 dB 2 s | 6 kHz 80 dB 2 s | Center Light 0.5 s |
| #2 | 100 Hz clicker 75 dB 1 s | white noise 75 dB 1 s | Center Light 1 s |
| #3 | 100 Hz clicker 75 dB 1 s | white noise 75 dB 1 s | Center Light 1 s |
| #4 | 100 Hz clicker 75 dB 1 s | white noise 75 dB 1 s | Center Light 1 s |
| #5 | 100 Hz clicker 75 dB 1 s | white noise 75 dB 1 s | Center Light 1 s |
| #6 | 100 Hz clicker 75 dB 1 s | white noise 75 dB 1 s | Center Light 1 s |
| #7 | 100 Hz clicker 75 dB 1 s | white noise 75 dB 1 s | Center Light 1 s |

**Table 2 | Timing of the three landmark sessions in each animal and the number of recorded BF neurons in each session**

Sessions (relative to each animal's $D_2$ session)

| Animal ID | #Bursting neurons / #BF neurons | $D_2-3$ | $D_2-2$ | $D_2-1$ | $D_2$ | $D_2+1$ | $D_2+2$ | $D_2+3$ | $D_2+4$ |
|---|---|---|---|---|---|---|---|---|---|
| #1 | | 4/13 | 1/4 | 23/32 | 25/33 | 34/49 | 28/41 | 31/46 | 29/42 |
| #2 | | 26/43 | 29/46 | 26/38 | 25/40 | 25/42 | 13/28 | 15/31 | 18/36 |
| #3 | | | 33/39 | 20/29 | 29/38 | 26/36 | 21/33 | 26/34 | 30/40 |
| #4 | | | 17/27 | 15/21 | 17/23 | 21/30 | 21/25 | 24/26 | |
| #5 | | | | 29/35 | 32/39 | 30/38 | 30/39 | 29/36 | 31/35 |
| #6 | | | | 17/30 | 16/29 | 20/37 | 16/28 | 18/29 | 20/27 |
| #7 | | | | | 23/28 | 20/22 | 11/13 | 19/23 | |

$D_0$ | $D_1$   session symbols

## Initial learning was characterized by the rapid emergence of reward-seeking behaviors and corresponding increases in BF activities

To validate the behavioral observations and understand the underlying neural dynamics, we recorded BF neuronal activity throughout the learning process (Fig. 3a) and used the consistent $S^{left}$ sound as the control stimulus to identify stable populations of BF bursting neurons (Figs. 3b and S2). A total of 1453 BF single units were recorded over 45 sessions ($N = 7$ rats), of which 70% (1013/1453) were classified as BF bursting neurons based on their stereotypical phasic response to the $S^{left}$ sound ($22.5 \pm 7.3$ neurons per session, mean ± std) (Fig. 3b). The population response of BF bursting neurons were highly consistent across animals (Fig. 3c), and remained remarkably stable in $S^{left}$ trials throughout the learning process (Fig. 3d). The inclusion of $S^{left}$ trials therefore allowed us to record from stable and representative populations of BF bursting neurons throughout the learning process, and to investigate how their activities dynamically evolved in other trial types during learning at single trial resolution (Fig. 3e).

We first applied this approach to understand the behavioral and neural dynamics in the $D_1$ session because all three types of rightward licks emerged in this session (Fig. 2c, middle panel). Detailed analysis of behavioral responses from a representative session (Fig. 3e) revealed that the three types of rightward licking behaviors emerged abruptly after a transition point (see Methods for definition). Rightward licking behaviors were mostly absent before the transition point, and rapidly switched to almost 100% licking after the transition point. This pattern of abrupt transition

was consistently observed across all animals (Fig. 4a1), while the behavioral performance and BF activities in $S^{left}$ sound trials remained relatively stable (Fig. S3).

At the neuronal level, there was a corresponding increase in the activity of BF bursting neurons that rapidly emerged after the transition in rightward reward-seeking behaviors (Figs. 3e and 4b1). This increase in BF activity was most prominent in the epoch before the trial outcome as animals approached the reward port (Fig. 3e2), but this activity was not consistently aligned with intervening behavioral events (Fig. S4). In contrast, in trials before the transition point, BF bursting neurons did not show similar activity increases in the corresponding time window after exiting the fixation port (Figs. 3e and 4b1). We will refer to the maximum BF activity in this window as the BF evaluation response (see Methods for definition) because it reflected animals' internal evaluation when no sensory stimuli were presented during this epoch and because it was not consistently associated with intervening behavioral events.

## The emergence and subsequent elimination of non-rewarded behaviors were mirrored by changes in BF activity

The stepwise learning model predicted that, as animals learned about the lick-right action as the first reward predictor, animals would initially engage in all three types of rightward licking behaviors because they all contained this reward predictor (Fig. 2e). We therefore examined the respective prevalence of the three types of rightward licking behaviors after the transition point in the $D_1$ session (Fig. 4a). All three types of rightward licking behaviors emerged immediately after

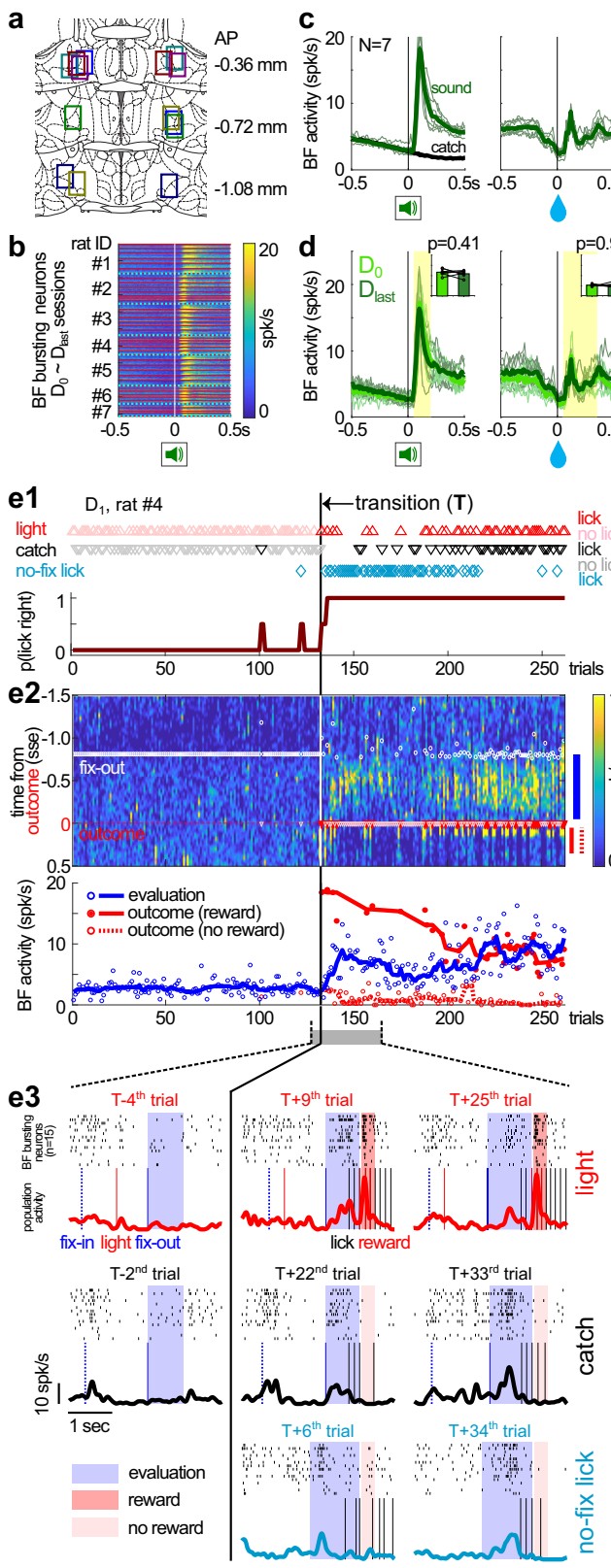

**Fig. 3 | Abrupt transition in reward-seeking behaviors corresponded to increased neuronal activity in the BF during initial learning. a**, Locations of electrode bundles targeting bilateral BF ($N = 7$) in coronal sections of the rat brain (coordinates relative to Bregma). Different colors correspond to different animals. Used with permission of Elsevier, from The rat brain in stereotaxic coordinates, Paxinos and Watson[34]; permission conveyed through Copyright Clearance Center, Inc. **b**, Response of individual BF bursting neurons ($n = 1013$) to the $S^{left}$ sound during new learning sessions ($N = 45$ sessions; separated by thin red lines) in each animal ($N = 7$ rats; separated by cyan dotted lines). BF bursting neurons showed robust and consistent phasic responses to the $S^{left}$ sound throughout the learning process. **c**, Average BF bursting neuron responses to $S^{left}$ sound onset and the associated reward delivery. BF activities in catch trials were plotted for comparison. Responses from individual animals (thin lines) were similar ($N = 7$). Symbols were adapted from Avila and Lin[22]. **d** The activity of BF bursting neurons remained stable between the first ($D_0$) and last ($D_{last}$) recording session. Average activities in the yellow shaded intervals were similar between these two sessions (inset) (two-sided paired $t$-test, $N = 7$). Thin lines indicate BF activity in individual animals. Symbols were adapted from Avila and Lin[22]. **e** Behavioral and BF neuronal dynamics in the $D_1$ session from a representative animal (rat #4). **e1** The emergence of three types of rightward licks after the transition point (top), and their combined rightward licking probability across trial types (bottom). The transition point (T) marked an abrupt transition in the pattern of reward-seeking behavior that went from no licking to 100% licking. **e2** Top, population activities of BF bursting neurons (color-coded) in the same trials (X-axis) as shown in **e1**. Y-axis indicates time in each trial, with time zero aligned at the trial outcome. No lick trials before the transition were aligned instead at the time of fixation port exit (fix-out, white circle) such that the median timing of fix-outs in lick and no lick trials were equivalent. The blue and red lines to the right of the panel indicate the time windows for calculating evaluation and outcome responses, respectively. Bottom, BF evaluation responses (blue) and outcome (red) responses across trials. Outcome responses were plotted separately for rewarded (solid red) and non-rewarded (dashed red) licks. Circles indicate BF activities in single trials and lines indicate their respective trends (moving medians). **e3** Examples of single trial BF activities from the three types of rightward licks taken around the transition point. Each panel showed the spike rasters of BF bursting neurons in this session ($n = 15$) (top), along with the population activity trace and relevant behavioral events (bottom). Shaded intervals indicate the time windows corresponding to evaluation responses (blue) and outcome responses (red) shown in **e2**. Notice that BF bursting neurons in the same session showed highly similar activity patterns, and that the BF evaluation response rapidly emerged in all three types of rightward licks after the transition point.

during the first learning step, which should be similarly present in all three types of rightward licking behaviors. Indeed, BF evaluation responses quickly increased after the transition point in all three types of rightward licking behaviors (Fig. 4b). The amplitudes of BF evaluation responses were similar between the three types of rightward licking behaviors within the first 60 trials after the transition. Subsequently, the BF evaluation response in no-fixation licks declined in the next 60 trials, relative to the other two types of rightward licking behaviors (Fig. 4b2–3).

These results support that the first step of the stepwise learning process corresponded to the first 60 trials after the transition point in the $D_1$ session. All three types of rightward licking behaviors were present during this step of learning. BF activity also increased to similar levels in all three types of behavioral sequences whenever animals approached and licked the right reward port, regardless of whether they had exited from the center fixation port.

These results further suggest that the second step of learning started at roughly 60 trials after the transition, at which point animals learned about the second reward predictor: exiting the fixation port (Fig. 2e). As a result, the no-fixation lick sequence was no longer compatible with the learned reward predictors, which resulted in diminished BF evaluation responses and the elimination of this behavior from the behavioral repertoire. The other two types of rightward licks, light licks and catch licks, remained compatible and maintained high levels of BF evaluation responses.

the transition point. Moreover, after about 60 trials, no-fixation licks began to decline and occurred less frequently than light licks and catch licks (Fig. 4a2–3).

At the neuronal level, the corresponding prediction was that the lick-right action would be associated with increases in BF activities

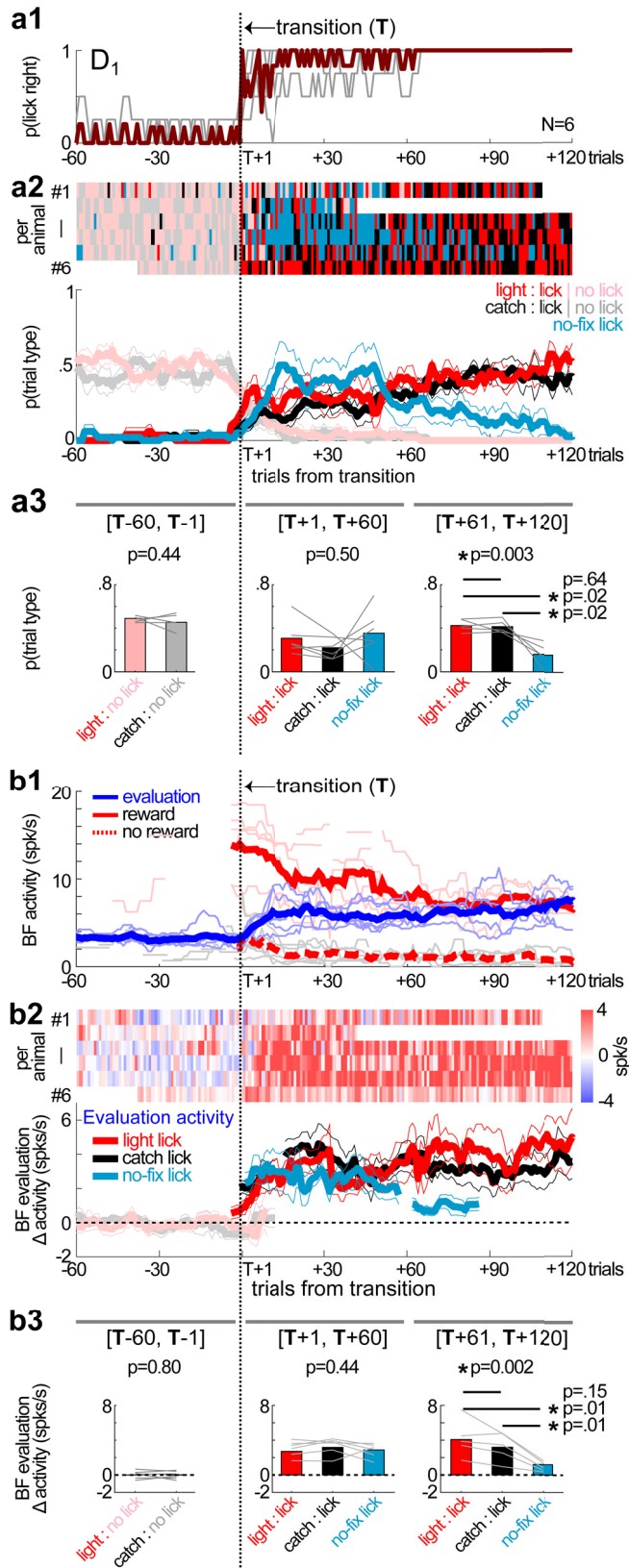

**Fig. 4 | Rapid emergence and subsequent refinement of reward-seeking behaviors and BF activity. a** The pattern of rightward licking behaviors aligned at the transition point in the $D_1$ session of each animal ($N = 6$). One animal (#7) with accelerated learning dynamics, in which its $D_1$ and $D_2$ occurred in the same session, was excluded from this analysis. The behavioral response patterns in three trial types (light trials, catch trials and no-fixation licks) were either pooled together within each animal (**a1**), or plotted separately (mean ± s.e.m.) (**a2**). While all three types of rightward licking emerged immediately after the transition, no-fixation licks subsequently decreased after 60 trials (**a3**) (Repeated measures ANOVA for group comparisons and post-hoc two-sided paired $t$-test between two trial types, $N = 6$). **b** Corresponding changes in BF activity aligned at the transition point in the $D_1$ session of each animal ($N = 6$). BF evaluation responses and outcome responses in the three trial types were pooled within each animal (**b1**), as in the example in Fig. 3e2. BF evaluation responses were further plotted separately for each trial type, relative to their respective baseline firing rates (mean ± s.e.m.) (**b2**). BF evaluation responses increased similarly in the three types of rightward licking immediately after the transition, and subsequently decreased in no-fixation licks after 60 trials (**b3**) (Repeated measures ANOVA for group comparisons and post-hoc two-sided paired $t$-test between two trial types, $N = 6$). Thin lines in **b1** indicate the trend (10-trial moving median) of BF activities from individual animals.

## BF neurons did not respond to the new light stimulus during initial learning

A further prediction of the stepwise learning model was that, at both the first and the second steps of learning, animals had not learned to use the new light stimulus as a reward predictor, and therefore the new light stimulus would not elicit responses in BF bursting neurons during these steps (Fig. 2e).

We tested this prediction by comparing BF activities between light and catch trials in the $D_1$ session (Fig. 5). Indeed, BF activities in light trials were highly similar to those in catch trials (in the absence of the light stimulus) in the epochs before exiting the fixation port. This was true regardless of whether animals subsequently licked at the right reward port. This observation confirmed the prediction that the new light stimulus did not activate BF bursting neurons in the $D_1$ session, despite the near-perfect behavioral performance in light trials after the behavior transition point.

On the other hand, in the epoch after exiting the fixation port, there were similar increases in BF activity when animals licked at the right reward port in both light and catch trials (Fig. 5). This increase in BF activity corresponded to the BF evaluation response described earlier (Fig. 4b), which reliably distinguished between lick and no lick trials, but not between light and catch trials (Fig. 5c). These observations support the idea that light and catch trials were treated as the same in the $D_1$ session, and that the light stimulus has not been learned as a reward predictor at this stage of learning.

## BF responses to light onset emerged later when the light stimulus was used to guide reward-seeking behavior

When did the third step of the stepwise learning process take place? Since the third step was when animals learned to use the light stimulus as a reward predictor to guide reward-seeking behaviors, it should correspond to the time point when the behavioral performance in light and catch trials began to diverge. We noted that the behavioral pattern in light and catch trials were highly similar prior to the $D_2$ session (Fig. 2c), and the similarity was best illustrated in the fine behavioral and neuronal dynamics within the $D_1$ session (Fig. 4). In contrast, during the $D_2$ session, the behavioral pattern in light and catch trials began to show a small but significant difference (Fig. 6a). This pattern suggests that the $D_2$ session might be when the third step of learning took place.

The corresponding prediction at the neuronal level was that BF bursting neurons should begin to respond to the light stimulus in the $D_2$ session. We tested this prediction by comparing BF activities between light and catch lick trials in the $D_2$ session, and found significant differences in all epochs, including the presence of a phasic response to the light onset (Fig. 6b). This observation supports the idea that BF responses to the new light emerged in the $D_2$ session.

To better understand how BF responses to the new light emerged in the D2 session, we further compared BF responses in light lick and catch lick trials between (1) late trials in the $D_2-1$ session; (2) early trials in the $D_2$ session; (3) late trials in the $D_2$ session (Fig. 6c). At the end of the $D_2-1$ session, BF neurons did not show response to the new light

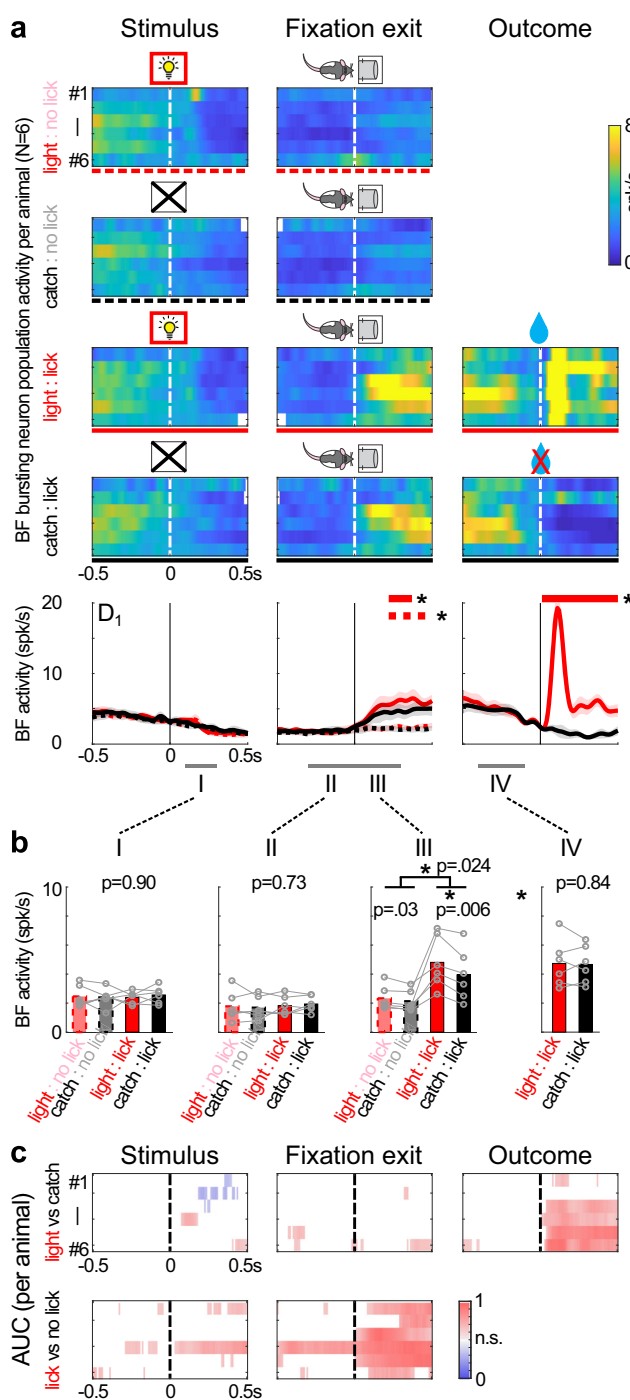

**Fig. 5 | BF neurons did not respond to the new light stimulus during initial learning. a** Population activities of BF bursting neurons in light and catch trials in the $D_1$ session. BF activities from individual animals ($N = 6$) (top) and group averages (mean ± s.e.m.) (bottom) were plotted separately based on trial type (light or catch) and behavioral response (licking at the right reward port or not), and aligned at three key behavioral events in each trial: stimulus onset, fixation port exit, and trial outcome. The solid and dashed lines below the top panels indicate the line symbols used in the bottom panels. Horizontal lines in bottom panels indicate significant differences in population BF activities between light lick and catch lick trials (solid red), or between lick and no lick trials (dashed red) (statistical significance defined as $p < 0.01$ for 3 consecutive steps using two-sided paired $t$-test for each 100 ms sliding window and 10 ms step). Symbols were adapted from Avila and Lin[22]. **b** Comparison of BF activities between the four trial combinations during four time windows (I-IV). There was no difference in BF activities before exiting the fixation port (I & II). In contrast, BF activities increased in both light and catch lick trials after exiting from the fixation port (III & IV). BF activities in light lick and catch lick trials were similar before the trial outcome (IV). (Repeated measures ANOVA for group comparisons and post-hoc two-sided paired $t$-test, $N = 6$). **c** Comparison of BF activities between light and catch trials (top) or between lick and no lick trials (bottom) using sliding window ROC analysis, aligned at the three behavioral events. Only significant ($p < 0.001$, two-sided) area-under-curve (AUC) values were shown. BF activities could not reliably discriminate between light and catch trials before the receipt of reward. In contrast, BF activities could reliably discriminate between lick and no lick trials after exiting the fixation port.

responses to the new light preceded the elimination of catch licks during the third step of learning (Fig. S5).

**Stronger BF responses to the new light reflected better learning and faster decisions**

After the third step of learning took place in the $D_2$ session, did the learning about the new light plateau or did the learning continue to progress? At the neuronal level, BF phasic responses to the new light continued to grow stronger after the $D_2$ session (Fig. 7a). At first glance, the continual increases in BF responses to the new light stimulus did not match with the hit rates in light trials, which had already plateaued in the $D_2$ session (Fig. 2c). However, as we have shown earlier (Figs. 4 and 5), hit rates in light trials could be a poor index of learning about the new light stimulus because light licks in the early learning sessions were not driven by the light stimulus but by later events in the behavioral sequence (exiting fixation and lick-right) as reward predictors. Those behavioral events enabled rightward licking responses in the absence of the light stimulus (catch licks and no-fixation licks).

Instead of the hit rate in light trials, a better behavioral index for the learning about the light stimulus would be the difference in the levels of behavioral performance between light and catch trials. This behavioral index accounted for the contributions from the later events in the behavioral sequence that were shared between light and catch licks, and isolated the contribution of the light stimulus to the rightward licking behavior. We found that this index of light learning was strongly correlated with the amplitude of BF phasic responses to the light stimulus in individual sessions (Fig. 7b). This observation therefore supports the idea that the learning about the light stimulus continued to grow stronger after the $D_2$ session.

Another dimension of the learning about the light stimulus was the change in animals' decision speeds, measured by reaction times (RTs). Previous studies have shown that stronger BF bursting responses are quantitatively coupled with, and causally lead to, faster RTs[21,22]. Such observations predicted that there would be corresponding decreases in RTs toward the light stimulus after the $D_2$ session. In support of this prediction, we found that stronger phasic bursting responses to light onset were coupled with faster RTs in individual sessions (Fig. 7c). Together, these observations support the idea that the learning about the new light was reflected by the amplitude of BF phasic response to the light stimulus.

(paired $t$-test between light lick and catch lick trials, $p = 0.51$, $N = 6$). In early $D_2$ trials, BF phasic responses to the new light were clearly visible in 4 animals. Comparison between late trials in the $D_2$−1 session and the early $D_2$ trials showed a trend toward increasing BF responses (Fig. 6c1, significant at $p < 0.05$ level; Fig. 6c2, paired $t$-test, $p = 0.065$, $N = 6$). Moreover, during the $D_2$ session, BF responses to the new light increased in all animals between early and late trials (Fig. 6c2, paired $t$-test, $p = 0.003$, $N = 7$). This increase took place mostly during the sustained phase of the BF response that was better aligned with fixation port exit than with stimulus onset (Fig. 6d). Together, these observations suggest that BF responses to the new light developed partly offline between the $D_2$−1 and $D_2$ session, and partly strengthened during the $D_2$ session. Comparing the temporal dynamics of behavioral and BF responses further suggests that the emergence of BF

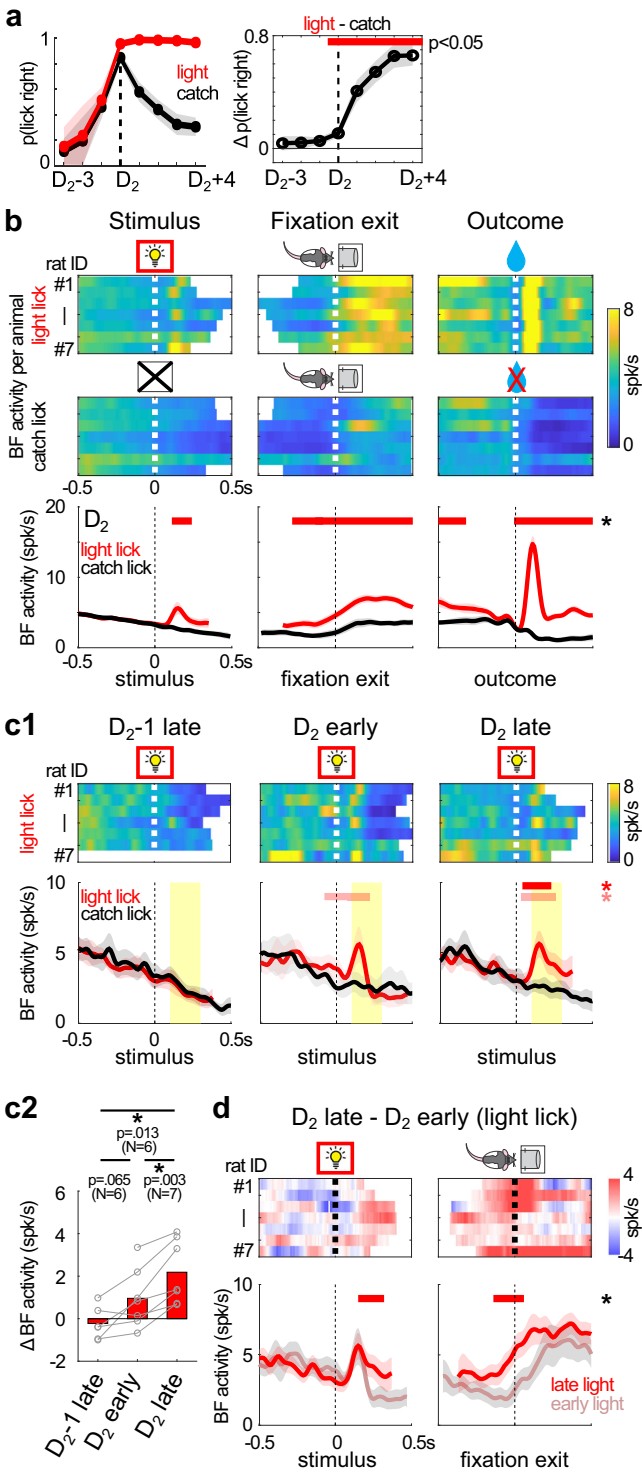

**Fig. 6 | BF responses to light onset emerged later when light was used to guide reward-seeking behavior. a** The probability of rightward lick in light and catch trials (mean ± s.e.m.), aligned to the $D_2$ session of each animal ($N = 7$) (left). Their difference, shown as mean ± s.e.m., began to differ significantly at the $D_2$ session (two-sided paired $t$-test, $p < 0.05$) (right). **b** Population activities of BF bursting neurons in light lick and catch lick trials in the $D_2$ session for individual animals (top) ($N = 7$) and their group average (mean ± s.e.m.) (bottom). BF activities in light lick trials were significantly higher in epochs before exiting the fixation port. Significant differences in BF activities (two-sided paired $t$-test, $p < 0.01$, 3 consecutive bins) were indicated by horizontal red lines. BF activities were truncated at the median RT of the respective sessions (see Methods for details). Symbols were adapted from Avila and Lin[22] . **c** The emergence of BF responses to the new light in the $D_2$ session. **c1** BF activities in light lick and catch lick trials at three points of the learning process: the last 20 trials (late) in the $D_2-1$ session ($N = 6$); first 20 trials (early) in the $D_2$ session ($N = 7$); late trials in the $D_2$ session ($N = 7$). Top row depicts BF activities in light lick trials from individual animals, while the bottom row depicts population BF activities (mean ± s.e.m.) in light lick and catch lick trials. Significant differences in BF activities between the two trial types were indicated by horizontal lines (two-sided paired $t$-test; red: $p < 0.01$, 3 consecutive bins; pink: $p < 0.05$, 3 consecutive bins). Yellow shaded intervals indicate the time windows for calculating BF responses to the new light in **c2**. **c2** Average BF responses to the new light, defined as the difference between the two trial types, at the three points of the learning process (two-sided paired $t$-test, $N$ indicated in the figure). **d** Comparison of BF activities between early and late light lick trials in the $D_2$ session, aligned at stimulus onset and fixation port exit. Top row shows the activity difference in individual animals, and the bottom row shows population BF activities in the two trial types (mean ± s.e.m.) ($N = 7$). Significant differences in BF activities (two-sided paired $t$-test, $p < 0.01$, 3 consecutive bins) were indicated by horizontal lines. The activity difference was stronger and better aligned at fixation port exit. Symbols were adapted from Avila and Lin[22].

## Reward expectations negatively modulated BF responses to trial outcomes

A final validation of the learning dynamics came from how BF responses to the trial outcome were modulated by reward expectations and BF activities in earlier epochs. Previous studies have found that the response of BF bursting neurons to the reward was negatively modulated by reward expectation[21,22,27,28]. Such properties would predict that, in the current experiment, BF responses to the reward should decrease throughout the stepwise learning process. Indeed, BF responses to the right reward decreased over trials in the $D_1$ session (Figs. 3e2 and 4b1), and continued to decrease over subsequent sessions (Fig. 7a). These results support that animals learned to better predict the rewarding outcome across different steps of the learning process.

We further investigated whether the response of BF bursting neurons to trial outcomes were negatively correlated with BF responses in earlier epochs. Such a negative correlation is a hallmark feature of reward prediction error encoding[29] and has been previously reported in BF bursting neurons[21,22]. Indeed, at the per session level, we found that the amplitude of BF responses to the reward in light trials was strongly and negatively correlated with the amplitude of BF phasic bursting response to the light onset (Fig. 8a). Moreover, in pre-$D_2$ sessions where BF responses to the new light stimulus had yet to develop, we found that BF responses to the trial outcome were negatively correlated with BF evaluation responses in the same trial (Fig. 8b). This negative correlation at the single trial level was observed not only when the reward was delivered (light licks) but also when the reward was absent (no-fixation licks and catch licks) (Fig. 8b, c). The fact that these patterns were observed in catch licks and non-fixation licks supports that animals were expecting to receive reward in those trials, and the extent of reward expectation was similarly reflected in BF evaluation responses and negatively modulated BF responses to the trial outcome. Together, these observations support that BF bursting neurons encoded reward prediction error, and that BF activities in

In addition, BF activities not only reflected the learning about the light stimulus, but also predicted reward-seeking behaviors in the absence of the light stimulus. For example, in catch trials, stronger BF activities after exiting the fixation port predicted rightward licking behavior and discriminated lick trials from no lick trials (Fig. S6a). Moreover, increased BF activities before the start of licking predicted longer durations of licking in catch lick trials when no reward was delivered (Fig. S6b). These observations provided additional support for the idea that the activity of BF bursting neurons promoted reward-seeking behaviors.

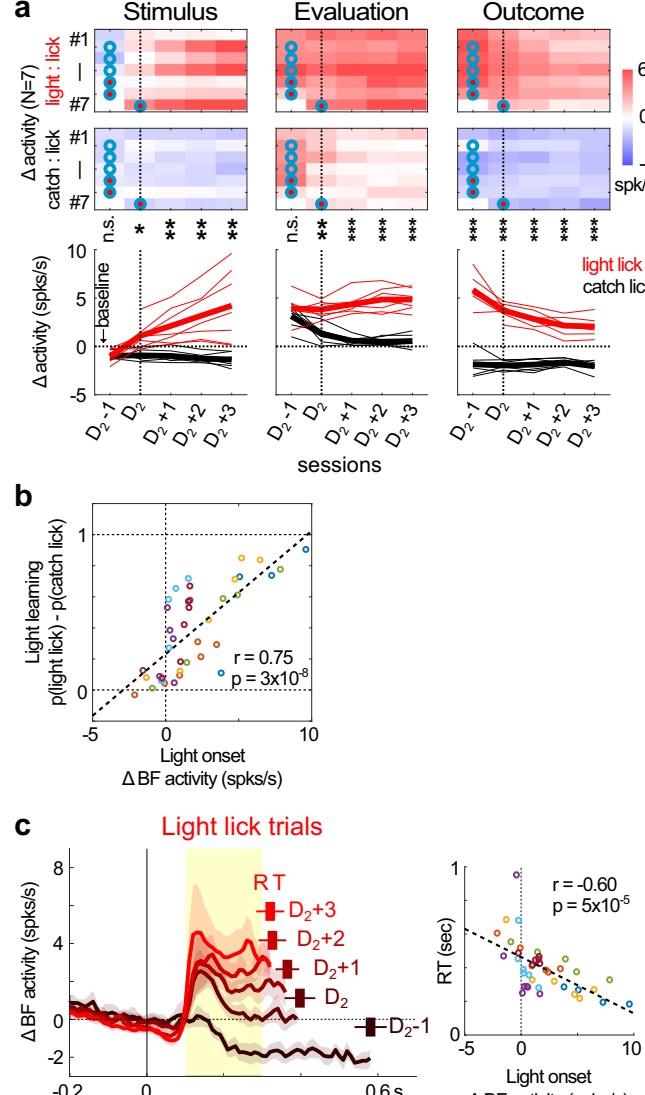

**Fig. 7 | Increased BF responses to light onset predicted the extent of light learning and faster reaction times. a** The evolution of BF activities in light and catch lick trials, aligned at each animal's $D_2$ session, calculated relative to their respective baseline firing rates. Significant differences in BF activities between the two trial types were indicated (two-sided paired $t$-test; *$p < 0.05$; **$p < 0.01$; ***$p < 0.001$). BF activities in the stimulus onset and evaluation epochs diverged in the $D_2$ session and their differences grew larger afterwards. Also notice that BF responses to the reward in light lick trials decreased over sessions. The red and cyan circles respectively indicate the $D_0$ and $D_1$ sessions in each animal, same as in Fig. 2c. **b** The average amplitude of BF responses to light onset in light lick trials was positively correlated (Pearson correlation) with the behavioral index of light learning, i.e. the difference in the probability of rightward licks between light and catch trials. BF activities were calculated relative to their respective baseline firing rates in each session. Each circle indicates one session and different colors correspond to different animals. **c** Average responses of BF bursting neurons (mean ± s.e.m.) to light onset in light lick trials (left), calculated relative to each animal's $D_2$ session. BF activities, relative to their respective baseline firing rates, were truncated at the median RT of the respective sessions. The RTs (mean ± s.e.m.) of the corresponding sessions were shown above each trace. Stronger BF responses to light onset in the [0.1, 0.3]s window (yellow shaded interval, left panel) were correlated (Pearson correlation) with faster RTs in individual sessions (right). Each circle indicates one session and different colors correspond to different animals, same as in **b**.

earlier epochs (stimulus onset or evaluation window) reflected animals' reward expectations.

## Discussion

Results from the current study support that animals used a stepwise strategy to learn behavioral sequences that contain both sensory stimuli and their own actions. Behavioral events were learned sequentially, starting from the event closest to the reward and sequentially expanded to earlier events (Fig. 1). The behavioral signature of this stepwise learning process was the sequential refinement of rightward licking behaviors, in which non-rewarded licking behaviors (no-fixation licks and catch licks) were sequentially eliminated while the rewarded behavior (light licks) was preserved (Fig. 2). Learning about each behavioral event as a new reward predictor was accompanied by the emergence of BF responses toward that event, which conveyed animals' reward prediction toward that event. Increased BF activities first emerged in the epoch before animals entered the reward port (Figs. 3e and 4b), while responses to the earlier event (light stimulus) developed later (Figs. 5 and 6). The evolution of BF activities mirrored the behavioral response patterns, which was initially increased in all three types of rightward licks (Figs. 3e and 4b) and subsequently decreased in non-rewarded licks as those behaviors were eliminated (Figs. 4b and 7a). Throughout the learning process, the activity of BF bursting neurons encoded reward prediction error signals (Fig. 8) and their increased activities consistently predicted reward-seeking behaviors and faster reaction times (Fig. 7). These results therefore identified the behavioral and neurophysiological signatures of the stepwise learning strategy when animals learned behavioral sequences.

The current study focused on the learning of behavioral sequences that contain both sensory stimuli and actions, which reflect behavioral contexts that are commonly encountered both in experimental and natural settings. The learning dynamics that we described in this study cannot be easily accounted for using either Pavlovian[1,2] or instrumental[3,4] conditioning alone. In this regard, stepwise learning provides a new framework to understand the learning of behavioral sequences.

Theoretically, long behavioral sequences are difficult to learn because the number of sequence permutations grows exponentially as a function of the sequence length. However, modeling studies have suggested that such learning can be greatly accelerated using the stepwise strategy, which reduces the number of sequence permutations to a linear function of the sequence length[18]. The current study provides behavioral and neurophysiological evidence to support that animals indeed adopt the stepwise learning strategy to learn behavioral sequences. At each step of learning, animals explored the subset of behavioral sequences that shared common sequence elements learned in previous steps (Figs. 1 and 2). Such explorations allowed animals to distinguish those sequences, which were initially indistinguishable at the beginning of that learning step, and selectively eliminate subsets of non-rewarded sequences. From this perspective, the stepwise learning strategy offers an intuitive explanation of why animals committed certain types of 'errors' (non-rewarded licks), and suggests that those behaviors in fact represented genuine reward-seeking efforts at earlier stages of learning. By learning the associative relationship one step at a time, the stepwise learning strategy likely minimized the cognitive burden of the animal during the learning process and enabled the efficient learning of complex sequences.

Our results suggest that stepwise learning is likely a general strategy for learning behavioral sequences because its behavioral and neural signatures were also observed in other learning settings. In a separate cohort of animals, we showed that the behavioral signatures of sequential refinements were similarly observed when the sensory modalities of the stimulus were reversed, and regardless of the laterality of the new learning side (Fig. S1). Moreover, even during the learning of the most simple behavioral sequence containing only two

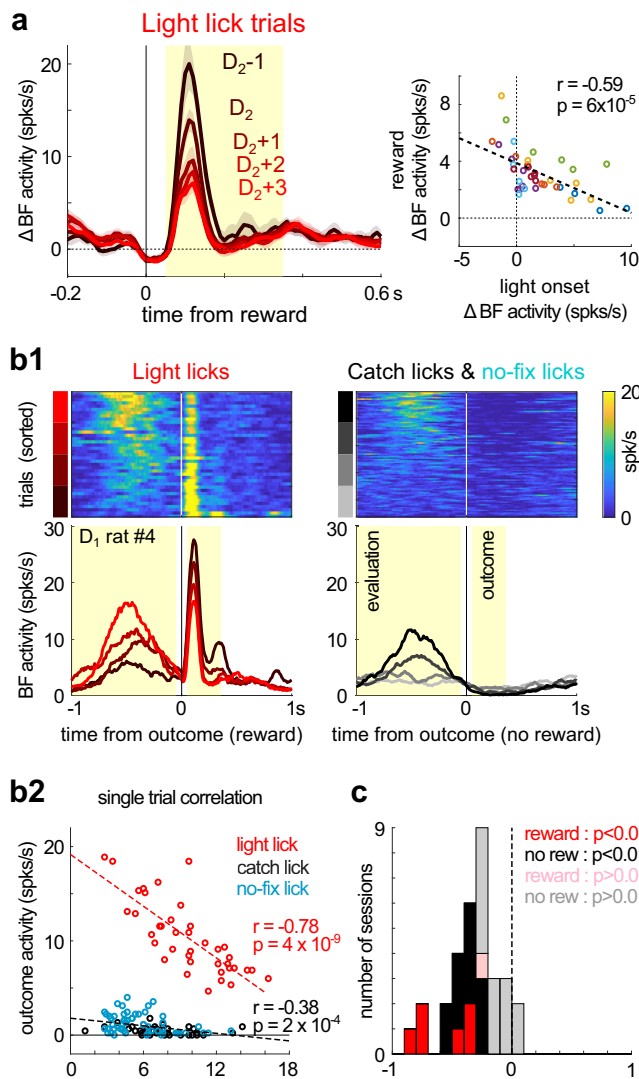

**Fig. 8 | BF responses to trial outcome were negatively correlated with BF activities in earlier epochs. a** Average responses of BF bursting neurons (mean ± s.e.m.) to the reward in light lick trials, relative to their respective baseline firing rates, plotted separately for the five sessions relative to the $D_2$ session (left). BF responses to the reward (yellow shaded interval, left panel) were negatively correlated (Pearson correlation) with BF responses to light onset (Fig. 7c) in individual sessions (right). Each circle indicates one session and different colors correspond to different animals. **b**, Negative correlation of BF activities between evaluation and outcome responses in single trials. **b1** Single trial BF activities in an example $D_1$ session, aligned at the trial outcome. Single trial BF responses were plotted separately for rewarded licks (light licks) and non-rewarded licks (catch licks and no-fixation licks). Yellow shaded intervals indicate time windows for calculating evaluation and outcome responses. Trials were sorted by the amplitude of evaluation responses (top). Average BF activities from the four quartiles of trials were plotted separately (bottom). **b2** Negative correlation (Pearson correlation) between single trial BF evaluation responses and outcome responses in this session. Each dot represents one trial. Catch licks and no-fixation licks were pooled together to calculate the correlation in non-rewarded licks. **c** Histogram of correlation coefficients (Pearson correlation) between evaluation and outcome responses from individual sessions. Results for rewarded lick trials (light licks) were calculated from pre-$D_2$ sessions when BF responses to the light stimulus had not developed ($N = 7$), and for non-rewarded lick trials from sessions with at least 50 trials of catch and no-fixation licks combined ($N = 25$). Most sessions showed significant negative correlations.

elements (stimulus-action), similar behavioral and neural signatures were also observed (Fig. S7).

It is important to note that, while these results support that stepwise learning is a widely-used strategy for learning behavioral sequences, they do not preclude other possible learning strategies. In particular, if animals had previously encountered similar behavioral events, such experiences could be generalized to the new learning context, and allow animals to use a forward chaining strategy to learn behavioral sequences, instead of the backward chaining order in the stepwise learning strategy[18]. For example, if animals had previously learned that light stimuli could predict reward, they could generalize that experience to the current learning task and view the new light stimulus as a potential reward predictor. Such generalizations would lead animals to engage in reward-seeking behaviors in light trials and quickly discover the light-reward association, bypassing the learning of behavioral events later in the sequence. Such strategies perhaps underlie the accelerated learning dynamics that we observed in one animal (animal #7, Fig. 2c), which also featured the strongest BF responses to the light stimulus during the initial encounter of the new light (Fig. 6b).

Our results revealed that the true temporal dynamics about the learning of the new stimulus can be very different from the dynamics of behavioral performance levels in the new stimulus trial. We found that, despite the near-perfect behavioral performance in light trials that began after the transition point in the $D_1$ session (Figs. 2 and 4), the learning of the new light stimulus did not begin until the $D_2$ session (Figs. 5 and 6). Moreover, the learning of the new light stimulus continued to grow after the $D_2$ session despite the plateaued behavioral performance in light trials (Figs. 6 and 7). During early stages of learning, while the light stimulus was clearly perceptible and had been consistently paired with reward over many trials, the reward-seeking behavior in light trials was not driven by the light stimulus but by later behavioral events in the sequence (fixation port exit and lick-right). The critical factors that allowed us to reach this conclusion were the analysis of non-rewarded licking responses and the inclusion of catch trials in our task design. If not for these factors, we would incorrectly conclude that the learning about the new light stimulus occurred much earlier.

The discrepancy between behavioral performance in light trials and the learning about the light stimulus highlights the potential mismatch in which salient physical events (such as the light stimulus) are not always automatically used by animals to predict reward, especially during the early stages of learning. The light stimulus was only incorporated as a reward predictor in later stages of the learning process, despite its continued presence from the beginning of the new learning. This observation indicates that, even in the absence of changes in reward contingencies in the environment, the learning process can lead to the addition of new reward predictors. Such structural revisions of the internal reward prediction model pose a fundamental challenge to theories and models of learning that assume a static set of reward predictors. Using models with incorrect reward predictors will lead to incorrect interpretations, regardless of how well the model fits the behavioral performance.

The current study extends our understanding about the roles of BF bursting neurons in the encoding of reward prediction error. Previous studies have demonstrated that BF responses to rewards are negatively modulated by reward expectation[21,22,27] and support the idea that BF neurons encode a reward prediction error signal[27,28]. The current study further extends this idea and shows that BF bursting neurons similarly encode reward prediction error in the context of new learning, and such encoding is robust even at the single trial level (Fig. 8). This robust encoding of reward prediction error by BF bursting neurons quantitatively conveys the amount of reward prediction associated with each behavioral event at each learning step. As a result, the stepwise learning process was mirrored by the activity of BF

bursting neurons, which provides a neural correlate of the stepwise learning process that we were able to track throughout the learning process in single trials. Given that reward prediction error information is also conveyed by other neuromodulatory systems, including midbrain dopaminergic neurons[12,13,29] and BF cholinergic neurons[30–32], BF bursting neurons are likely part of a broader network with partly correlated activities. Whether the other systems also provide similar neural correlates of stepwise learning remains to be determined.

Previous studies have also established that BF bursting neurons serve as a bidirectional gain modulation mechanism for reward-seeking behaviors, where increased BF activities promote faster reaction times[21,22] while the inhibition of BF activities leads to rapid behavioral stopping[23]. Manipulations of BF activities using electrical stimulation further suggest that BF bursting neurons likely play a causal role that can modulate decision speed when their activities are increased[22] or inhibited[23]. The current study extends this idea to the context of new learning, by showing that increased BF activities were tightly coupled with reward-seeking behaviors at multiple levels throughout the learning process (Figs. 3–7), and quantitatively predicted faster reaction times (Fig. 7) and longer licking durations (Fig. S6). The engagement in reward-seeking behaviors was particularly important in the context of learning behavioral sequences, because such explorations were essential for discovering the relationship between earlier events in the behavioral sequence and the rewarding outcome. Taken together, BF bursting neurons may serve the role of transforming the encoding of reward-prediction error into promoting reward-seeking behaviors during the new learning process.

Finally, several studies have suggested that BF bursting neurons are likely a subset of GABAergic neurons[21,24,33], but their specific cellular marker(s) remains to be determined. Such marker information will be needed to conduct selective manipulation experiments that specifically target BF bursting neurons to directly test their causal role in the learning process.

## Methods

### Ethics statement
All experimental procedures were conducted in accordance with the National Institutes of Health (NIH) Guide for Care and Use of Laboratory Animals and approved by the National Institute on Aging (NIA) Animal Care and Use Committee and by the Institutional Animal Care and Use Committee at the National Yang Ming Chiao Tung University, Taiwan (NYCU).

### Subjects
Seven male Long-Evans rats (Charles River, NC), aged 3–6 months and weighing 300–400 g were used for the recording experiment. Rats were housed in a 12/12 day/night cycle and were provided with 10–12 dry pellets per day and unrestricted access to water. Rats were trained in daily sessions lasting 60–90 min. A separate cohort of eight male Long Evans rats (National Laboratory Animal Center, Taiwan) were used for behavioral testing (Fig. S1) and an additional recording experiment (Fig. S7). During training and recording procedures, these rats were water restricted to their 85–90% weight and were trained in a daily 60-min session. Water-restricted rats received 15 min water access at the end of each training day with free access on weekends.

### Apparatus
Plexiglass operant chambers (11″ L × 8 ¼″ W × 13″ H), custom-built by Med Associates Inc. (St. Albans, VT), were contained in sound-attenuating cubicles (ENV-018MD) each with an exhaust fan that helped mask external noise. Each chamber's front panel was equipped with an illuminated nose-poke port (ENV-114M) located in the center (horizontal axis) as the fixation port, which was equipped with an infrared (IR) sensor to detect the entry of the animals' snout into the port. On each side of the center nose-poke port there were two reward ports (CT-ENV-251L-P). Two IR sensors were positioned to detect reward-port entry and sipper-tube licking, respectively.

Sucrose solution (13.3%) was used as reward and delivered through the sipper tubes located in the reward ports. Reward delivery was controlled by solenoid valves (Parker Hannifin Corp #003-0111-900, Hollis, NH) and calibrated to provide 10 μl of solution per drop. Each chamber was equipped with a ceiling-mounted speaker (ENV-224BM) to deliver auditory stimuli, and a stimulus light (ENV-221) positioned above the center fixation port to serve as the new light stimulus. For the additional behavioral testing (Fig. S1), one stimulus light each was added above the left and the right reward ports to serve as the sensory cue in the visual discrimination experiment, and water was used as the reward. Behavioral training protocols were controlled by Med-PC software (Versions IV & V, Med Associates Inc.), which stored all event timestamps at 1 or 2 ms resolution and also sent out TTL signals to the neurophysiology recording systems.

### Behavioral training procedures
Rats were trained in operant chambers that were dimly lit. Rats were first trained in an auditory or visual discrimination task. See Table 1 and Fig. S1 for details of the stimuli used for each animal. Trials were separated by an unsignaled inter-trial interval (ITI) lasting 4–6 s. Fixation or licks during the ITI reset the ITI timer. After the ITI, the center fixation port was illuminated, which was turned off only when rats poked the fixation port. Rats were required to maintain fixation in the center nose-poke port for a variable amount of foreperiod. Four different foreperiods (0.35, 0.5, 0.65, and 0.8 s) were used, pseudorandomly across trials. After the foreperiod, one of three conditions was randomly presented with equal probabilities: a $S^{right}$ stimulus indicated reward on the right port; a different $S^{left}$ stimulus indicated reward on the left port; or the absence of stimulus (catch trial) indicated no reward. An internal timestamp was recorded in catch trials to mark the onset of the would-be stimulus. Early fixation port exit before the end of the foreperiod led to the re-illumination of the center fixation port. Licking in the correct port within a 3 s window after stimulus onset led to three drops of reward, delivered starting at the 3rd lick. The delivery of reward at the 3rd lick created an expectation for trial outcome at this time point, which was dissociated in time from the initiation of licking (1st lick). We will therefore refer to the time point of the 3rd lick also as the trial outcome event. Trials with licking responses ended after 1 s following the last lick, while trials without licking responses ended after the 3 s response window. The ending of each trial started the ITI timer.

After reaching asymptotic behavioral performance in the auditory or the visual discrimination task, a new learning phase was introduced by replacing either the $S^{right}$ or $S^{left}$ stimuli by a novel sensory stimulus in a different sensory modality, while all other aspects of the task remained the same. In the group that were initially trained with auditory discrimination, the $S^{right}$ sound stimulus was replaced by the central light above the fixation port to indicate reward on the right port (Table 1). In the group there were initially trained with visual discrimination, either the $S^{right}$ or $S^{left}$ light stimuli was replaced by a 6 kHz sound (70 dB) played from a speaker above the center fixation port (Fig. S1).

### Stereotaxic surgery and electrode
Surgery was performed under isoflurane anesthesia similar to our earlier study[22]. Multiple skull screws were inserted to anchor the implant, with one screw over the cerebellum serving as the common electrical reference and a separate screw over the opposite cerebellum hemisphere serving as the electrical ground. Craniotomies were opened to target bilateral BF (AP −0.6 mm, ML ± 2.25 mm relative to Bregma)[34]. The electrode contained two bundles of 16 polyimide-insulated tungsten wires (38 μm diameter; California Fine Wire, CA), each bundle ensheathed in a 28-gauge stainless steel cannula and

controlled by a precision microdrive. The impedance of individual wire was ~ 0.1 MΩ measured at 1 kHz (niPOD, NeuroNexusTech, MI or Open Ephys Acquisition Board). During surgery, the cannulae were lowered to DV 6.5 mm below cortical surface using a micropositioner (Model 2662, David Kopf Instrument or Robot Stereotaxic, Neurostar GmbH) at a speed of 2–50 μm/s. After reaching target depth, the electrode and screws were covered with dental cement (Hygenic Denture Resin), and electrodes further advanced to 7.5 mm below the cortical surface. Rats received ibuprofen and topical antibiotics after surgery for pain relief and prevention of infection, and were allowed one week to recover with *ad libitum* food and water. Cannulae and electrode tip locations were verified with cresyl violet staining of histological sections at the end of the experiment. All electrodes were found at expected positions between AP [−0.2, −1.2] mm, ML [1.5, 3] mm, relative to Bregma, and DV [7.5, 8.5] mm relative to cortical surface (Fig. 3a).

### Data acquisition and spike sorting
Electrical signals were referenced to a common skull screw placed over the cerebellum. Electrical signals were filtered (0.3 Hz to 7.5 kHz) and amplified using Cereplex M digital headstages and recorded using a Neural Signal Processor (Blackrock Microsystems, UT). Single unit activity was further filtered (250 Hz to 5 kHz) and recorded at 30 kHz. Spike waveforms were sorted offline to identify single units using the KlustaKwik sorting algorithm followed by a custom Python GUI (version 2.7) for manual curation. Only single units with clear separation from the noise cluster and with minimal (<0.1%) spike collisions (spikes with less than 1.5 ms interspike interval) were used for further analyses, consistent with previous studies of BF bursting neurons[21–26]. Additional cross-correlation analysis was used to remove duplicate units recorded simultaneously across multiple electrodes[21–26].

### Recording during the new learning phase
After surgery, BF neuronal activity was monitored while rats were re-trained in the auditory discrimination task to asymptotic performance level. During this re-training phase, BF electrode depths were adjusted slightly (by advancing electrodes at 125 μm increment) until a stable population of BF single units can be recorded. At this point, the new learning phase with the light as the new $S^{right}$ stimulus was introduced and rats were trained and recorded daily with BF electrodes remained at the same depth. This approach allowed us to monitor the activity of a large population of BF neurons and follow its temporal evolution across sessions.

### Data analysis
Data were analyzed using custom Matlab (R2018b, MATLAB The MathWorks Inc., Natick, MA) scripts.

**Define different behavioral response types.** Licking responses were defined for stimulus ($S^{left}$ and $S^{right}$) and catch trials if rats licked at least three times in the reward port within the 3 s window after stimulus onset (or the corresponding timestamp for the would-be stimulus in catch trials). Licking responses to the correct reward port were rewarded with three drops of water, delivered starting at the 3rd lick (referred to as trial outcome event). During the new learning phase, licking responses in the new stimulus and catch trials were predominantly to the reward port associated with the new stimulus.

No-fixation licks corresponded to licking responses to the reward port associated with the new stimulus that were not preceded by poking the center fixation port. Specifically, no-fixation licks were defined based on three criteria: (1) rats made at least three consecutive licks in the reward port; (2) the interval between the last exit from the fixation port and the first lick must be greater than 2 s; (3) the interval between the last exit from the reward port and the subsequent first lick in the same reward port must be greater than 1 s. These duration thresholds were determined based on the empirical licking patterns

across animals. In the analyses of learning dynamics in the $D_1$ session (Figs. 3 and 4), no-fixation licks were treated as rightward licking trials, even though such behaviors were self-initiated and not imposed by the task design.

Reaction time (RT) in light trials in a session (Fig. 7c) was defined as the median of the interval between the onset of the light stimulus and the exit from the fixation port in light lick trials. Lick duration in catch trials in a session (Fig. S6) was defined as the median of the interval between the first and the last lick in catch lick trials.

**Define the $D_0$, $D_1$ and $D_2$ learning landmarks.** During the new learning phase, three sessions ($D_0$, $D_1$, $D_2$) were identified in individual animals as landmarks that demarcated distinct stages of new learning (Fig. 2c). The $D_0$ session was defined as the very first session the new light stimulus was introduced. The $D_1$ session was defined as the first session when animals began to respond correctly in the new stimulus trials and obtained reward in the associated reward port in at least three trials. The $D_2$ session was defined as the session in which catch licks occurred most frequently. The $D_1$ and $D_2$ landmarks allowed us to identify similar learning stages across animals despite their individual differences in learning dynamics. The specific timing of the three landmark sessions in each animal are provided in Table 2 (also see Fig. 2c). One animal (ID#7) with accelerated learning dynamics, in which $D_1$ and $D_2$ occurred in the same session, was excluded from analyses of $D_1$ neural dynamics (Figs. 4 and 5) to ensure that neural activities associated with $D_2$ did not confound the neural dynamics in the $D_1$ session. The BF neuronal activity in this animal was included in the analysis of $D_2$ neural dynamics (Fig. 6) and showed the strongest phasic response to the light onset among all animals, consistent with its accelerated learning dynamics.

**Identification of the behavioral transition point in the $D_1$ session.** The behavioral transition points in $D_1$ sessions (Figs. 3 and 4) were identified based on behavioral response patterns in three trial types combined: light trials, catch trials and no-fixation licks. The behavioral response pattern in each trial was coded as either 1 or 0 based on whether animals licked in the right reward port in that trial. The behavioral transition point was defined as the point with the largest difference in licking responses between the 20 trials before and the 20 trials after that point. In 5/7 animals, the first trial after the behavioral transition was a rewarded light lick trial. In the other two animals, the transition point was adjusted to the closest light lick trial by 2 or 4 trials, respectively.

**Identification of BF bursting neurons.** BF bursting neurons were defined as BF single units whose average firing rates during the [0.05, 0.2]s window after stimulus onset increased by more than 2 spikes/s in the $S^{left}$ sound trials compared to the corresponding window in catch trials (Fig. S2). This contrast between sound trials and catch trials was necessary because it removed the nonstationary baseline before stimulus onset and allowed us to ask whether BF neurons truly responded to the sound stimulus. In addition, BF bursting neurons should have baseline firing rates (during the [−1, 0]s window relative to the trial start signal) less than 10 spikes/s.

A total of 1453 BF single units were recorded over 45 sessions ($N$ = 7 rats), of which 70% (1013/1453) were classified as BF bursting neurons based on their stereotypical phasic response to the $S^{left}$ sound (22.5 ± 7.3 neurons per session, mean ± std) (Fig. S2 and Table 2). Since BF electrodes remained at the same depth throughout the recording sessions, the same BF single units might be recorded in multiple sessions. The large number of BF bursting neurons recorded in each session allowed us to treat them as a representative sample of all BF bursting neurons, whose responses to the $S^{left}$ sound were highly stable throughout the learning process (Fig. 3b-d). This strategy ensured that we were following functionally the same neuronal ensemble and could

track how BF bursting neurons acquired responses to the new light during learning, regardless of whether the identities of these BF neurons were exactly the same in each session. One session with only one BF bursting neuron was excluded from the analysis of BF population activities.

**Population BF responses to behavioral events**. The spike timestamps of all BF bursting neurons in a single session were pooled together to approximate the population activity of all BF bursting neurons. Population peri-stimulus time histograms (PSTHs) were calculated with 10-ms bins, and normalized by the number of BF bursting neurons in a session.

To properly assess whether BF bursting neurons responded to the onset of the new light stimulus (Figs. 5–7), it was important to disambiguate such stimulus-onset responses from the increased BF activities after fixation port exit (Fig. 5). To achieve this goal, PSTHs to the stimulus onset were calculated based only on spikes that occurred before fixation port exit in individual trials, resulting in different interval lengths (between stimulus onset to fixation port exit) across trials. Accordingly, the calculation of the mean PSTH across all trials in a session was adjusted for the different number of trials at different interval lengths. The mean PSTHs were further truncated at the median interval length of that session to reduce noisy estimates of PSTHs at long interval lengths due to lower number of trials. When PSTHs were averaged across animals, the averaged PSTHs were further truncated at the mean of median interval latencies across animals. This truncation procedure resulted in the uneven lengths of PSTHs across individual animals (Figs. 5, 6, S4 and S5) and across sessions (Fig. 7). This procedure was also applied to calculating the BF responses before fixation port exit to include only spikes that occurred after stimulus onset (Figs. 5, 6 and S4), and for calculating BF responses during licking in catch lick trials (Fig. S6).

The time windows used to quantify average BF activity in different epochs were indicated in respective figures, and corresponded to the following: [0.05, 0.2]s after $S^{left}$ sound onset; [0.1, 0.3]s after $S^{right}$ light stimulus onset; [0.1, 0.3]s after the timestamp for the would-be stimulus in catch trials; [0.05, 0.35]s after the 3rd lick for outcome responses; [−0.3, 0]s and [0, 0.3]s relative to the fixation port exit. The epoch for calculating evaluation response is described below.

Evaluation response (Figs. 3e, 4b, 7a and 8b, c) refers to the increased BF activity after exiting the fixation port and before the trial outcome (3rd lick). The evaluation response reflected animals' internal evaluation because no additional sensory stimuli were presented during this epoch and this activity was not consistently aligned with intervening behavioral events (Fig. S4). Specifically, the evaluation response was calculated in individual trials and defined as the maximum firing rate of any 500 ms window during the evaluation epoch, which corresponded to the interval between [fix-out, outcome], with additional adjustments according to trial types. In light lick and catch lick trials, the evaluation epoch was defined as [fix-out, outcome] in each trial. The epoch durations in light lick and catch lick trials within each session were used as the reference point for other trial types as described next. In no-fixation licks, in which the fix-out event was absent, the duration of evaluation epoch was set as the 95th percentile of the evaluation epoch durations in light lick and catch lick trials, and the epoch should begin at least 0.5 s before reward port entry. In light no lick and catch no lick trials, in which the 3rd lick event was absent, the duration of the evaluation epoch was set as the median of the evaluation epoch durations in light lick and catch lick trials. These adjustments in the definition of evaluation epochs, as well as its calculation of maximum firing rate within the epoch, took into consideration the behavioral variability across trial types, learning stages and individual animals.

To evaluate the dynamic changes of BF activities around the transition point in the $D_1$ session (Figs. 3e and 4b), single trial

evaluation and outcome responses were smoothed using moving median over 10 trials. The smoothed trends were aligned at the transition point and then averaged across all animals. Only trials with smoothed trend data from at least 4 animals were plotted in the group average (Fig. 4b).

**Statistics**. Statistical comparisons were conducted using the Statistics and Machine Learning Toolbox (version 11.3) in MATLAB (R2018a) (https://www.mathworks.com/). Two-sided paired $t$-test (ttest.m) was used to compare behavioral and neural activity differences between two groups (Figs. 3d, 6a, 6c2, and 7a). Repeated measures analysis of variance (ranova.m) was used for comparisons involving more than 2 groups, by specifying the appropriate within-subject models (Figs. 4a3, 4b3, and 5b). Comparisons of PSTHs between two groups (Figs. 5a, 6b, c1, d, and S3c) was conducted for each 100 ms sliding window (10 ms step) using two-sided paired $t$-test. Significance level was set at $p < 0.01$ for three consecutive bins. Pearson correlation (corrcoef.m) was used to determine the relationship between neuronal activities and/or behavior (Figs. 7b, c, 8a, b2, c and S6b).

**Receiver operating characteristic (ROC) and area under curve (AUC) analysis**. To determine whether the activity of BF bursting neurons differentiated between trial types within each $D_1$ session (Fig. 5c), we compared BF activity for each 100 ms sliding window (10 ms step) using the AUC measure of ROC analysis (auc.m by Alois Schloegl). At each sliding window, BF population activity was calculated for each light and catch trial, and distributions of BF activities were compared between light vs catch trials or between lick vs no lick trials. Significance level was set at $p < 0.001$ using 10,000 trial-shuffled random permutations (two-sided).

To determine whether BF activity differentiated between lick and no lick trials within the same trial type (light or catch trials) (Fig. S6a), we compared BF activity in the [0, 500] ms window after exiting the fixation port. For each session and each trial type, lick and no lick trials must each constitute at least 10% of that trial type to be included in the analysis. Catch trials from all sessions were included in this analysis. Only light trials before the $D_2$ session (pre-$D_2$) were included in this analysis because BF responses to the onset of the light stimulus had not developed in those sessions. Significance level was set at $p < 0.05$ using 1000 trial-shuffled random permutations (two-sided).

**Reporting summary**
Further information on research design is available in the Nature Portfolio Reporting Summary linked to this article.

## Data availability
Source data and statistics used to make each figure are provided with this paper. The raw datasets analyzed in the current study are available upon reasonable request to the authors. Source data are provided with this paper.

## Code availability
Data analysis was performed by built-in and custom Matlab scripts and are available from the corresponding author upon request.

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

## Acknowledgements

We thank P.R. Rapp, A. Scaglione, S.W. Wu, K.L. Hsieh and M.C. Lee for critical discussions of the manuscript; B.M. Brock and S.T. Wu for technical support. This research was funded by the Intramural Research Program of the National Institute on Aging (NIH, USA) (1ZIAAG000339) and by NARSAD Young Investigator Award to S.L. Additional supports came from the National Science and Technology Council (NSTC, Taiwan) grants (107-2320-B-010-028-MY3; 108-2638-B-010-002-MY2; 110-2628-B-A49A-505) to S.L., and from the Brain Research Center, National Yang Ming Chiao Tung University from The Featured Areas Research Center Program within the framework of the Higher Education Sprout Project by the Ministry of Education (MOE) in Taiwan to S.L.

## Author contributions

H.E.M. and S.L. designed the study. H.E.M., K.V., Y.J. and H.C. performed experiments and collected data. H.E.M. and S.L. analyzed data. H.E.M. and S.L. wrote the manuscript with inputs from K.V., Y.J., and H.C.

## Competing interests

The authors declare no competing interests.
