## [Peer Review File · Nature Communications]

The behavioral signature of stepwise learning strategy in male rats and its neural correlate in the basal forebrainREVIEWER COMMENTS

Reviewer #1 (Remarks to the Author):

In this study, Manzur et al. presented a compelling case for the 'stepwise learning' model, in which an animal is associating reward with progressively earlier events (including both stimuli and actions) according to an event sequence during a learning task. Moreover, basal forebrain bursting neurons showed activation patterns that correlated with the behavioral signatures of stepwise learning, exhibited hallmarks of reward prediction error coding and predicted behavioral performance.

This study required a careful, well-controlled behavioral design; but even with such paradigms and protocols, capturing the fine details of behavioral learning is a hard task that requires skillful and insightful analyses, likely being a major reason for the scarcity of such studies and a consequential gap in our understanding. Filling this gap is paramount if we aim to better understand the neural mechanisms of associative learning and I think this study makes a major inroad into this important problem, making a significant conceptual advance. I have a few comments; since these are mostly either points of clarifications or suggestions to increase readability, they can largely be considered as only minor concerns.

1. Understanding (and memorizing) the sequence of behavioral events (and elimination of unrewarded sequences) is key to follow and appreciate this manuscript; therefore, I feel it would greatly help the reader to explicitly list the important steps, something like

1st step: learned: right lick -> reward

2nd step: learned: fix-out + right lick -> reward; eliminated: no-fix licks

3rd step: learned: light + fix-out + right lick -> reward; eliminated: catch licks

2. The activity changes of BF neurons are correlated with updates in the behavioral strategies. Do the data provide any hints on one or the other (i.e., behavioral learning vs. emergence of BF response) happening earlier? For instance, can a direct comparison be made between learning of light as reward predictor / elimination of catch licks and BF burst responses to light? Could some form of session-wise behavior – firing rate cross-correlations provide insight? When 'no-fix licks' and 'evaluation responses' for 'no-fix licks' decrease, does one of those happen earlier/faster?

3. The present study aligns well with papers stressing the role of dopaminergic RPE coding in the service of associative learning, providing a full-fledged theoretical framework (Kim, ..., Uchida) and demonstrating signs of progressively earlier DAergic activation during learning (Amo, ..., Watabe-Uchida,

2022). However, this theoretical framework has recently been challenged, by proposing a competing 'causal learning' model instead (Jeong,...,Namboodiri, 2022). Does the present study speak to this debate, i.e., are there specific points that argue for one or the other model? If so, it could also be interesting to add this as a discussion point.

4. The motivation for focusing on the BF bursting neurons is provided at the end of the Introduction. While well-justified, it would be an important discussion point whether this population can be considered unique in its role in sequential learning, or likely part of a broader network with partly correlated activities.

5. In the first Results section, it is unclear what the Authors mean by 'behavioral sequences that contained all the learned events in that step would predict reward and therefore preferentially executed'; it would be better to rephrase this.

6. The introduction of BF 'evaluation response' on p. 6 is somewhat unclear; a more detailed explanation of the rationale would make this part easier to read.

7. 'BF activities in light trials were highly similar to those in catch trials (in the absence of the light stimulus), in the epochs after stimulus onset...' I am assuming that after 'hypothetical' stimulus onset in case of catch trials? Please clarify.

8. Discussion, p.10-p.11, 'observed in a subset of animal': observed in one animal.

9. Fig.3D – color difference of the thin lines is not visible; E1 legend: 'pattern OF reward seeking'. Fig.4A3 legend: more explanation needed. What statistics were used? Fig.5A last row, are the statistics from ROC analysis here?

10. In Fig.6C, it is hard to see the faint line corresponding to the 'early' trials. It looks like there is already a response that is similar in magnitude, but less sustained. Can the Authors comment on this? Also, is there any insight about the significant FR decrease for 'outcomes' (no reward) in catch lick trials? Could this be a RPE-like omission response? Is it absent in 'catch trial:no lick' trials?

11. In Fig.7A, what are the blue circles (with or without red filling)? Fig.7C legend should indicate that the activity was separated for different sessions relative to D2.

12. Fig.8C – What is 'lreward'? (Typo?)

13. Methods: if the number of recorded BF units is simply added up from all recording sessions, it probably contains neurons multiple times, as the electrodes were fixed during the training, right? Since the recorded cells are considered as a proxy for the BF bursting population, this does not really influence the results, but should be noted for clarity.

14. In Fig.S3A, what is in the last histogram?

I believe that openness and transparency can increase the fairness of peer review. Therefore, I decided to sign my reviews.

Balazs Hangya

Reviewer #2 (Remarks to the Author):

Summary

In this work, the authors substantively advance the case that neurons in the basal forebrain* signal reinforcement prediction error by recording from rats across the acquisition of a two alternative forced choice task. The pattern of changes, with respect to behavior and neurophysiological responses, evidence a process of learning the meaning of more temporal proximal events prior to more temporally distal events. It is demonstrated behaviorally that rats learn to eliminate non-rewarded lick behavior consequent to the use of a new policy-predictive cue (here a visual cue indicating reward available following right port entry), and that as the predictive visual cue's significance is learned, that the reaction time to that cue decreases. As rats learn that the preceding light cue indicates what reward seeking action to take, BF responses to predicted reward decrease, whereas BF responses to action, and then to the reward-predicting visual cue increases. This relationship, observed at the session level, is also demonstrated to hold on a per-trial level, and relate to the behavior and reaction time emitted by the animal.

The work is of significance in that it deepens the fields understanding of the role of BF in reporting reinforcement prediction signaling, which respects the progression through a sequence involving cues & actions linking these neurophysiological signatures to the course and manner of learning, and to behavioral performance.

I have no concerns regarding the collection and analysis of the data.

Below, I have but minor comments which the authors may consider as they see fit. In addition, I do have a few additional questions about what was/wasn't included, which could lead to additions that may further the authors' case.

*Specifically, functionally defined "bursting" neurons, that are presumptively but perhaps not exclusively GABA-ergic

Minor comments

Introduction

I do not personally find the discourse and contrasting of Pavlovian, instrumental conditioning, and the claim that what was done here is different than instrumental conditioning compelling or needed in establishing the significance of this work. The second half of intro is strong and recommend eliminating or reworking the pavlovian v instrument v sequence narrative.

Results

Sequential refinement of reward-seeking behaviors during new learning

-Is BF activity starting on fix out? Or on Side port entry? I recommend indicating when the side port entry occurs, and potentially adding analysis recentered in this event. Perhaps the authors have already done so, and finding that it did not more greatly account for across trial variability in response, analyzed with respect to fix-out. If not in the revision, please consider adding an assessment in the response.

Initial learning was characterized by the rapid emergence of reward-seeking behaviors and corresponding increases in BF activities

-What about auditory cued trials in which subject then sought reward at the light-port, and therefore wasn't rewarded but licked?

-What does responding to auditory cued trials look like, and is it disrupted at all, behaviorally or neurophysiologically, upon introducing the light cue?

-Shouldnt the left hand side of E2 show what the response profile was to the auditory-go right situation prior to transitioning? Rather than showing no-lick trials?

-What do the non-burst neurons do, and why are they not being shown in contrast to the burst population?

Behavior

--Since there are four different foreperiods in the fixation port prior to a stimulus being delivered (or not in the case of catch), what was the activity of BF neurons as those periods expired without a stimulus, if any?

-When in a catch trial, did rats wait typically to a time past the longest foreperiod before answering exiting and entering a side port?

-Isn't there a fourth type of right lick? Where it is an auditory cued trial, but the animal gets it incorrect by licking right? In those cases, what happens? Or are there too few as these trials?

The emergence and subsequent elimination of non-rewarded behaviors were mirrored by changes in BF activity

Fig 4a

-Figure 4 should include a column aligned to side port (right port) entry. My question is whether BF activation may be more precisely related to side port entry than to fixation port exit, and indicate the state of being in the side port.

BF neurons did not respond to the new light stimulus during initial learning

Fig5a

-Here too Figure 5 should include a column aligned to side port (right port) entry. My sense is that BF activation may be more related to side port entry than to fixation port exit, and indicate the state of being in the side port.

BF responses to light onset emerged later when the light stimulus was used to guide reward-seeking behavior

-6c left: I'm surprised from the data shown in 6c that early light v early catch does not show a significant difference. Can the authors confirm? It appears as if it should be different, with the duration of the response extending in the later light trials over that of the early light trials. In any case, the absence of light responses in D1 indicate that light-evoked responses emerge as light acquires predictive power.

-The suppression of the reward outcome-elicited response, and the augmentation to the reward predictive light stimuli are consonant with RPE, but how is elevated BF activity over the 'evaluation' period a sign of reward prediction error? Does the reward come at a random delay once in the side port? If so, then yes, could be a sign of RPE over that interval. Hmm- it doesn't come at a random delay, but rather on the occurrence of the 3rd lick. What is the distribution of the 3rd lick times across the session? This may then form a probability density function of reward delays from fix-out (or maybe better side port in) experienced by the animal, giving rise under a RPE framework for why there is a BF response across the 'evaluation' period.

Stronger BF responses to the new light reflected better learning and faster decisions
“plateaued” change to “plateau”

Citations

Additional papers assisting in the case that BF conveys reinforcement prediction error

(Hegedüs et al. 2023; Chubykin et al. 2013; Liu et al. 2015)

Chubykin, Alexander A., Emma B. Roach, Mark F. Bear, and Marshall G. Hussain Shuler. 2013. “A Cholinergic Mechanism for Reward Timing within Primary Visual Cortex.” *Neuron* 77 (4): 723–35.

Hegedüs, Panna, Katalin Sviatkó, Bálint Király, Sergio Martínez-Bellver, and Balázs Hangya. 2023. “Cholinergic Activity Reflects Reward Expectations and Predicts Behavioral Responses.” *iScience* 26 (1): 105814.

Liu, C. H., J. E. Coleman, H. Davoudi, K. Zhang, and M. G. Hussain Shuler. 2015. “Selective Activation of a Putative Reinforcement Signal Conditions Cued Interval Timing in Primary Visual Cortex.” *Current Biology: CB* 2015/05/26 (12): 1551–61.

RESPONSE TO REVIEWERS' COMMENTS

Reviewer #1 (Remarks to the Author):

In this study, Manzur et al. presented a compelling case for the 'stepwise learning' model, in which an animal is associating reward with progressively earlier events (including both stimuli and actions) according to an event sequence during a learning task. Moreover, basal forebrain bursting neurons showed activation patterns that correlated with the behavioral signatures of stepwise learning, exhibited hallmarks of reward prediction error coding and predicted behavioral performance.

This study required a careful, well-controlled behavioral design; but even with such paradigms and protocols, capturing the fine details of behavioral learning is a hard task that requires skillful and insightful analyses, likely being a major reason for the scarcity of such studies and a consequential gap in our understanding. Filling this gap is paramount if we aim to better understand the neural mechanisms of associative learning and I think this study makes a major inroad into this important problem, making a significant conceptual advance. I have a few comments; since these are mostly either points of clarifications or suggestions to increase readability, they can largely be considered as only minor concerns.

Thank you! We are glad that the reviewer appreciated our findings and conceptual advances. In the following, we provide a point-by-point response to your comments and questions. In particular, we have made substantial progress on two issues raised by the reviewer: (1) how BF bursting neurons develop responses to the new light in the D₂ session; (2) the temporal relationship between the emergence of BF bursting neurons to the new light and the elimination of catch lick responses.

1. Understanding (and memorizing) the sequence of behavioral events (and elimination of unrewarded sequences) is key to follow and appreciate this manuscript; therefore, I feel it would greatly help the reader to explicitly list the important steps, something like

1st step: learned: right lick -> reward

2nd step: learned: fix-out + right lick -> reward; eliminated: no-fix licks

3rd step: learned: light + fix-out + right lick -> reward; eliminated: catch licks

Great suggestion! We have modified Figure 2E as the following.

Figure 2E. A stepwise learning model that accounts for the sequential refinement of the three types of rightward licking behaviors. **E1**, The behavioral events in the model arranged in the format as in Figure 1B. **E2**, Behavioral sequences learned as reward predictors at the three discrete steps of learning, along with the compatible and incompatible behavioral sequences at each step. **E3**, Sequential refinement of the three types of rightward licking behaviors arranged in the format as in Figure 1B.

2. The activity changes of BF neurons are correlated with updates in the behavioral strategies. Do the data provide any hints on one or the other (i.e., behavioral learning vs. emergence of BF response) happening earlier? For instance, can a direct comparison be made between learning of light as reward predictor / elimination of catch licks and BF burst responses to light? Could some form of session-wise behavior – firing rate cross-correlations provide insight? When ‘no-fix licks’ and ‘evaluation responses’ for ‘no-fix licks’ decrease, does one of those happen earlier/faster?

Thank you for the very insightful suggestion. Per your suggestion, we have conducted additional analyses and now report that the emergence of BF responses to the new light preceded the elimination of catch licks (**New Figure S5**, please see next page).

To determine the temporal order between those two events, we used cumulative sums to visualize the temporal dynamics of BF responses to the new light (red) and the divergence of behavioral responses between light and catch trials (blue). We designed the cumulative sum measures to contrast the difference between light and catch trials, such that the lack of BF responses to the new light or the lack of behavioral response differences between light and catch trials would result in a zero-slope line. Thus, the timing for the emergence of BF or behavioral responses can be visualized as the divergence point of cumulative sum traces away from the horizontal line.

At the neuronal level, the lack of BF responses to the new light, which was evident in the D_2-1 sessions, resulted in zero-slope cumulative sums. On the other hand, the presence of BF responses to the new light resulted in cumulative sums with positive slopes, which first emerged in D_2 sessions (red arrows) and remained positive afterwards. At the behavioral level, the lack of behavioral response differences between light and catch trials, which was evident in the D_2-1 sessions, resulted in zero-slope cumulative sums. On the other hand, the elimination of catch licks resulted in cumulative sums with positive slopes, which emerged between D_2 to D_2+2 sessions in different animals (blue arrows) and remained positive afterwards.

Note that the timing of the emergence (red and blue arrows) were visually determined based on when the cumulative sums diverged from the horizontal line ($y=0$). We were not able to apply a universal criterion to rigorously define this timing because of individual differences in learning dynamics. Nevertheless, the overall pattern supports that the emergence of BF responses to the new light (all in D_2 sessions) occurred earlier than the elimination of catch licks (between D_2 to D_2+2 sessions) in all animals. These results are now added as the new supplementary Figure S5.

The second question about no-fix licks was not further pursued because of the behavioral variabilities across animals in the D_1 session, and that no-fix licks were behaviorally less constrained and therefore their relationship with BF activities were harder to quantify.

Figure S5. The emergence of BF responses to the new light preceded the elimination of catch licks.

A, An example D₂ session showing how we used cumulative sums to visualize the temporal dynamics of BF responses to the new light (red) and the divergence of behavioral responses between light and catch trials (blue), plotted over the number of trials in the session (light and catch trials combined). The corresponding single-trial behavioral responses and BF activities are plotted below. BF response to the new light in each trial was defined as the difference between the average activity in the [0.1, 0.3]s window after light stimulus onset and the average activity in the corresponding window in all catch trials in the same session. The absence of BF responses to the new light would result in zero-slope cumulative sums. To determine the timing for the elimination of catch licks, we calculated the difference between the cumulative sums of the rightward licking numbers (normalized by the number of each trial type in the session) between light and catch trials. The lack of behavioral response differences between light and catch trials would result in a

zero-slope line. Thus, the timing for the emergence of BF or behavioral responses can be visualized as the divergence point of cumulative sum traces away from the horizontal line. In this example session, BF responses to the new light emerged earlier than the elimination of catch licks, indicated respectively by the red and blue arrow. **B**, Each panel represents cumulative sums from one animal (each row) and one session (each column, relative to the D_2 session). To account for individual variabilities in BF response amplitudes across animals, the cumulative sums of BF responses were scaled relative to the maximum of the cumulative sum in the D_2+2 session in each animal. The increasing slopes of the cumulative sums between D_2 to D_2+2 sessions reflected the increasing response amplitudes to the new light (as shown in Figure 7A). The timing of the emergence (red and blue arrows) were visually determined based on when the cumulative sums diverged from the horizontal line ($y=0$). **C**, The average BF activity in light lick and catch lick trials in the corresponding sessions (relative to the D_2 session). The time window used to calculate BF responses to the new light is highlighted in yellow.

3. The present study aligns well with papers stressing the role of dopaminergic RPE coding in the service of associative learning, providing a full-fledged theoretical framework (Kim, ..., Uchida) and demonstrating signs of progressively earlier DAergic activation during learning (Amo, ..., Watabe-Uchida, 2022). However, this theoretical framework has recently been challenged, by proposing a competing ‘causal learning’ model instead (Jeong, ..., Namboodiri, 2022). Does the present study speak to this debate, i.e., are there specific points that argue for one or the other model? If so, it could also be interesting to add this as a discussion point.

This is an important developing debate in this field, but we do not think the current study can address this argument. The dopaminergic RPE framework has been an important guiding inspiration for this study, and our descriptions of RPE encoding in the BF are generally consistent with this framework. On the other hand, the new ‘causal learning’ idea proposed by Jeong et al sought to provide a more inclusive framework that is compatible with the classical findings in the dopamine literature, as well as the new experimental results that contradict the predictions of the RPE framework. Based on our own experience studying the relationship between BF activity and reward-seeking behaviors, we suspect that the clues to reconcile the debate between RPE framework and causal learning might reside in the details of the behavioral experiments.

4. The motivation for focusing on the BF bursting neurons is provided at the end of the Introduction. While well-justified, it would be an important discussion point whether this population can be considered unique in its role in sequential learning, or likely part of a broader network with partly correlated activities.

Thank you for the question. We do not believe that BF bursting neurons are unique in the encoding of RPE. We now add the following to the discussion section: “Given that reward prediction error information is also conveyed by other neuromodulatory systems, including midbrain dopaminergic neurons and BF cholinergic neurons, BF bursting neurons are likely part of a broader network with partly correlated activities. Whether the other systems also provide similar neural correlates of stepwise learning remains to be determined.”

5. In the first Results section, it is unclear what the Authors mean by 'behavioral sequences that contained all the learned events in that step would predict reward and therefore preferentially executed'; it would be better to rephrase this.

We have rephrased as the following: As more behavioral events are learned as reward predictors in each step, only behavioral sequences that contain all the learned events would predict reward and therefore preferentially executed, while incompatible sequences that do not contain all the learned events would not predict reward and therefore be eliminated from the behavioral repertoire.

6. The introduction of BF 'evaluation response' on p. 6 is somewhat unclear; a more detailed explanation of the rationale would make this part easier to read.

Thank you for the suggestion. We have provided additional analyses in response to a related concern raised by Reviewer#2 to show that the 'evaluation response' was not consistently associated with any behavioral event (including fixation port exit, side port entry and the first lick) (Please see **Figure S4, Figures R1-R2**). This was the reason why the 'evaluation response' was defined as the maximum BF activity in the epoch after exiting the fixation port and before licking. We called this the 'evaluation response' to convey the idea that it occurred in the absence of external sensory stimuli and was not consistently associated with any behavioral events, and likely reflected animals' internal evaluation. We have modified the text on p.6 accordingly.

In the revision process, we also modified the evaluation epoch definition in no-fix lick trials by adding an additional requirement to ensure that the evaluation epoch begins at least 0.5s before reward port entry in those trials. This change was necessary because we noted that the evaluation epoch that we used (which was set as the 95th percentile of the evaluation epoch durations in light lick and catch lick trials) was not long enough in a small number of no-fix lick trials to cover the reward port entry event. This modification is now indicated in Methods, which also resulted in slight changes to Figures 3, 4 and 8.

7. 'BF activities in light trials were highly similar to those in catch trials (in the absence of the light stimulus), in the epochs after stimulus onset...' I am assuming that after 'hypothetical stimulus onset in case of catch trials? Please clarify.

Yes you are correct. We have removed the confusing description 'after stimulus onset' in this sentence.

8. Discussion, p.10-p.11, 'observed in a subset of animal': observed in one animal.

Thank you. We have edited the text accordingly.

9. Fig.3D – color difference of the thin lines is not visible; E1 legend: 'pattern OF reward seeking'. Fig.4A3 legend: more explanation needed. What statistics were used? Fig.5A last row, are the statistics from ROC analysis here?

We have edited Figure 3D and legends in Figures 3E1, 4A3 and 5A accordingly.

Figures 4A3 & 4B3: add (Repeated measures ANOVA for group comparisons and post-hoc paired t-test between two trial types, N=6)

Figure 5A: add (statistical significance defined as $p < 0.01$ for 3 consecutive steps using paired t-test for each 100 ms sliding window and 10ms step).

10. In Fig.6C, it is hard to see the faint line corresponding to the 'early' trials. It looks like there is already a response that is similar in magnitude, but less sustained. Can the Authors comment on this? Also, is there any insight about the significant FR decrease for 'outcomes' (no reward) in catch lick trials? Could this be a RPE-like omission response? Is it absent in 'catch trial:no lick' trials?

Thank you for the insightful questions. We have modified Figure 6C and provided additional analyses to better understand how BF responses to the new light emerged in the D_2 session. We first compared BF responses in light lick and catch lick trials between three parts of the learning curve: (1) late trials in the D_{2-1} session; (2) early trials in the D_2 session; (3) late trials in the D_2 session. At the end of the D_{2-1} session, BF neurons did not show response to the new light (paired t-test between light lick and catch lick trials, $p = 0.51$, $n = 6$). In early D_2 trials, BF phasic responses to the new light were clearly visible in 4 animals. Comparison between late trials in the D_{2-1} session and the early D_2 trials showed that there was a trend toward increasing BF responses (Figure 6C1, significant at $p < 0.05$ level; Figure 6C2, paired t-test, $p = 0.065$, $N = 6$). These observations suggest that the BF phasic response to the new light could develop offline in between sessions. In addition, during the D_2 session, BF responses to the new light increased in all animals between early and late trials (Figure 6C2, paired t-test, $p = 0.003$, $N = 7$). This increase took place mostly during the sustained phase of the BF response that was better aligned with fixation port exit than with stimulus onset (Figure 6D). Together, these new results extend our observation in Figure 6B that BF responses to the new light developed in the D_2 session by suggesting that BF responses to the new light developed partly offline (between D_{2-1} and D_2 session), and partly strengthened during the D_2 session.

Figure 6C. The emergence of BF responses to the new light in the D₂ session.

C1, BF activities in light lick and catch lick trials at three points of the learning process: the last 20 trials (late) in the D₂-1 session; first 20 trials (early) in the D₂ session; late trials in the D₂ session. Top row depicts BF activities in light lick trials from individual animals, while the bottom row depicts population BF activities (mean ± s.e.m.) in light lick and catch lick trials (N=6-7). Significant differences in BF activities between the two trial types were indicated by horizontal lines (red: p<0.01, 3 consecutive bins; pink: p<0.05, 3 consecutive bins). Yellow shaded intervals indicate the time windows for calculating BF responses to the new light in C2. **C2**, Average BF responses to the new light, defined as the difference between the two trial types, at the three points of the learning process (paired t-test, N=6-7). **D**, Comparison of BF activities between early and late light lick trials in the D₂ session, aligned at stimulus onset and fixation port exit. Top row shows the activity difference in individual animals, and the bottom row shows population BF activities in the two trial types (mean ± s.e.m.) (N=7). Significant differences in BF activities (p<0.01, 3 consecutive bins) were indicated by horizontal lines. The activity difference was stronger and better aligned at fixation port exit.

The reviewer's second comment about BF Inhibition in catch lick trials during the outcome epoch was indeed similar to the RPE-like omission response. This inhibition in the outcome epoch was present in both catch lick and no-fix lick trials (documented in Figure 8B and 8C), and the activity during this outcome epoch was negatively correlated with the amplitude of preceding BF evaluation responses (Figure 8B, 8C), consistent with dynamics of reward prediction error encoding. No inhibition was observed in catch:no lick trials, as can be visualized from the example in Figure 3E2, particularly in trials before the transition point.

11. In Fig.7A, what are the blue circles (with or without red filling)? Fig.7C legend should indicate that the activity was separated for different sessions relative to D2.

The red and cyan circles in Figure 7A respectively indicate the D₀ and D₁ sessions in each animal, as in Figure 2C (aligned at D₂). Figure legends for 7A and 7C have been edited accordingly.

12. Fig.8C – What is 'reward'? (Typo?)

Good catch! Thank you! It should be 'reward'.

13. Methods: if the number of recorded BF units is simply added up from all recording sessions, it probably contains neurons multiple times, as the electrodes were fixed during the training, right? Since the recorded cells are considered as a proxy for the BF bursting population, this does not really influence the results, but should be noted for clarity.

The reviewer is correct. We have further clarified this in the method by adding: "Since BF electrodes remained at the same depth throughout the recording sessions, the same BF single units might be recorded in multiple sessions."

14. In Fig.S3A, what is in the last histogram?

Sorry for the confusion. We have modified this figure and added legend to show that this panel indicates the proportion of statistically significant sessions in light or catch trials from the previous panel. Note that this figure is now re-labeled as Figure S6 due to the addition of new supplementary figures.

I believe that openness and transparency can increase the fairness of peer review. Therefore, I decided to sign my reviews.

Balazs Hangya

Thank you for the detailed and insightful review. We are lucky to have a knowledgeable and constructive reviewer like you.

Reviewer #2 (Remarks to the Author):

Summary

In this work, the authors substantively advance the case that neurons in the basal forebrain* signal reinforcement prediction error by recording from rats across the acquisition of a two alternative forced choice task. The pattern of changes, with respect to behavior and neurophysiological responses, evidence a process of learning the meaning of more temporal proximal events prior to more temporally distal events. It is demonstrated behaviorally that rats learn to eliminate non-rewarded lick behavior consequent to the use of a new policy-predictive cue (here a visual cue indicating reward available following right port entry), and that as the predictive visual cue's significance is learned, that the reaction time to that cue decreases. As rats learn that the preceding light cue indicates what reward seeking action to take, BF responses to predicted reward decrease, whereas BF responses to action, and then to the reward-predicting visual cue increases. This relationship, observed at the session level, is also demonstrated to hold on a per-trial level, and relate to the behavior and reaction time emitted by the animal.

The work is of significance in that it deepens the fields understanding of the role of BF in reporting reinforcement prediction signaling, which respects the progression through a sequence involving cues & actions linking these neurophysiological signatures to the course and manner of learning, and to behavioral performance.

I have no concerns regarding the collection and analysis of the data.

Below, I have but minor comments which the authors may consider as they see fit. In addition, I do have a few additional questions about what was/wasn't included, which could lead to additions that may further the authors' case.

Thank you for appreciating the importance of this work. In the responses below, we address your constructive feedback with additional analyses and figures. In particular, we provide new analyses to address three key issues: (1) whether BF activity was aligned at the side reward port entry; (2) characterize the behavioral and neural dynamics for the well-learned sound trials during the new learning phase; (3) characterize how long animals waited in the fixation port in catch trials and the associated BF neural activity.

*Specifically, functionally defined "bursting" neurons, that are presumptively but perhaps not exclusively GABA-ergic

Minor comments

Introduction

I do not personally find the discourse and contrasting of Pavlovian, instrumental conditioning, and the claim that what was done here is different than instrumental conditioning compelling or needed in establishing the significance of this work. The second half of intro is strong and recommend eliminating or reworking the pavlovian v instrument v sequence narrative.

Thank you for the suggestion and for approving our rationale of studying behavioral sequence learning. The inclusion of Pavlovian and instrumental conditioning in the introduction was meant to draw the interest of a broader audience who may not be familiar with the idea of behavioral sequence learning. Most of our test audience have found this framing helpful.

Results

Sequential refinement of reward-seeking behaviors during new learning

-Is BF activity starting on fix out? Or on Side port entry? I recommend indicating when the side port entry occurs, and potentially adding analysis recentered in this event. Perhaps the authors have already done so, and finding that it did not more greatly account for across trial variability in response, analyzed with respect to fix-out. If not in the revision, please consider adding an assessment in the response.

Thank you for raising this important question about whether BF activity might be better aligned with side reward port entry. In the figures below, we re-plot the results from Figures 3-6 and aligned BF activity to intervening behavioral events (including fixation port exit, side reward port entry and the first lick). These analyses show that, while BF activity increased in all three types of rightward licks, the timing of increased BF activity varied substantially across trials and trial types, as well as across sessions and animals. No single event was able to capture the full dynamics of BF activities during new learning. This was the main reason why we chose to define 'evaluation response' as the maximum BF activity in the epoch after exiting the fixation port and before licking, which showed better correspondence with behavioral patterns than BF activities aligned at a single behavioral event.

(1) New Figure S4: Variability in the timing of peak BF activities across animals and sessions

We replotted population BF activities in light lick and catch lick trials during D_1 (from Figure 5A) and D_2 (from Figure 6B) sessions and aligned at two additional events (reward port entry and first lick). While the average population activity across all animals peaked around the reward port entry event (Figure S4, lower panels of A & B), BF activity patterns in individual animals (top panels) varied substantially. In particular, the maximum firing rate occurred at different time points in different animals, and also varied between the two sessions. Therefore, it was not feasible to use a single fixed time window around any single behavioral event to capture the rich dynamics of BF activity during the learning process.

Figure S4. Variability in the timing of peak BF activities across animals and sessions

Population activities of BF bursting neurons in light lick and catch lick trials in the D₁ session (from Figure 5A) (A) and D₂ session (from Figure 6B) (B) aligned at five different events: stimulus onset, fixation exit, right reward port entry, first lick and outcome. The maximum firing rate occurred at different time points in different animals, and also varied between the two sessions. Conventions as in Figure 5A.

(2) Reviewer-only Figure R1: BF activity from the example D₁ session (Figure 3E2) was not aligned at one specific behavioral event

To investigate the variability of BF activities at the single trial level, we replotted BF activity from Figure 3E2 and aligned at different behavioral events (Figure R1). Notice that the relative timing between different behavioral events within the same trial showed substantial variability in trials right after the transition point (Figure R1B). The variability decreased subsequently and the behavioral pattern became more stereotypical in later trials. At the neural activity level, there was no clear indication that BF activities were best aligned with one particular event over the others (Figure R1A).

Figure R1. BF activity from the example D_1 session was not aligned at one specific behavioral event

A, BF activity from the example D_1 session in Figure 3E2, aligned in each row at a different event: fixation exit, right reward port entry, first lick, outcome. The timing of all behavioral events were overlaid on the PSTHs. **B**, The same behavioral events (color-coded) were plotted alone for comparison. Conventions as in Figure 3E2.

(3) Reviewer-only Figure R2: Compare BF evaluation responses with BF activities aligned at reward port entry

In Figure R2, we compared the population evaluation response from Figure 4B with BF activities aligned at reward port entry. While these two measures of BF activities showed similar temporal patterns and increased after the transition point (Figure R2A), these two measures differed when BF activities were classified based on the three rightward licking trial types, specifically in no-fix licks (Figure R2B & C). While the BF evaluation response was not different between the three rightward licking trial types in the first 60 trials after the transition (Figure R2B), BF activities aligned at reward port entry were significantly lower in no-fix licks than the other two trial types (Figure R2C). This underestimation of BF activities in no-fix licks likely reflected the increased behavioral variability right after the transition point (Figure R1B).

Figure R2. Compare BF evaluation responses with BF activities aligned at reward port entry in the D_1 session

A, BF evaluation responses aligned at the transition point in the D_1 session (from Figure 4B1) were plotted along with BF activities aligned at reward port entry. BF activities were averaged in the [-0.25 0.25]s window around reward port entry, which was chosen to match the peak BF activity in Figure S4. **B-C**, The same BF activities from panel A, plotted separately for the three rightward licking trial types (as in Figure 4B2). Unlike BF evaluation responses (**B**), BF activities aligned at reward port entry (**C**) were significantly lower in no-fix licks than the other two trial types in the first 60 trials after the transition.

Together, these observations suggest that using a fixed time window aligned at reward port entry (or any specific behavioral event) would underestimate the increase in BF activities, especially in no-fix licks where the behavior was most variable. These observations were the reasons why we chose the maximum BF activity in characterizing the dynamics of the learning process. We called this the ‘evaluation response’ to convey the idea that it occurred in the absence of external sensory stimuli and was not consistently associated with any

behavioral events, and likely reflected animals' internal evaluation. We have now modified the text on p.6 to better convey this idea.

In the revision process, we also noted that the evaluation epoch duration that we used for no-fix licks (which was set as the 95th percentile of the evaluation epoch durations in light lick and catch lick trials) was not long enough in a small number of no-fix lick trials to cover the reward port entry event. As a result, we modified the evaluation epoch definition in no-fix lick trials by adding an additional requirement to ensure that the evaluation epoch begins at least 0.5s before reward port entry. This modification is now indicated in Methods, which also resulted in slight changes to Figures 3, 4 and 8.

Initial learning was characterized by the rapid emergence of reward-seeking behaviors and corresponding increases in BF activities

-What about auditory cued trials in which subject then sought reward at the light-port, and therefore wasn't rewarded but licked?

Thank you for the important question. We have conducted new analyses on sound trials during the D₁ session to characterize the behavioral and neurophysiological response patterns and added this to the revised manuscript as a supplementary figure.

New Figure S3: Behavioral and BF response patterns in sound trials during the D₁ session

We first compared behavioral response patterns in sound trials, catch trials and light trials during the D₁ session, aligned at the behavioral transition point that we identified in Figure 4 (Figure S3A). As expected, catch and light trials underwent abrupt behavioral transitions (similar to Figure 4A2, but each trial type was analyzed separately here). This abrupt behavioral transition also led to increased error rates (rightward licking) in sound trials, from $2.4 \pm 5.5\%$ (mean \pm std, N=7 rats) in pre-transition trials to $12.0 \pm 8.4\%$ in post-transition trials. While this increase in error rates was substantial, behavioral performance was still correct in the vast majority of post-transition sound trials in the D₁ session. For comparison, the overall error rate in sound trials across sessions was $6.4 \pm 3.3\%$ (mean \pm std, N=7 rats) (from Figure 2C).

In terms of neural activity, we found that BF response patterns to sound onset and reward in correct (leftward licking) trials remained fairly stable throughout the D₁ session. However, BF activity was substantially reduced in error trials when rats incorrectly licked at the right reward port (**Figure S3B, S3C**). These observations were confirmed at the population level across all animals (N=7): BF activity in correct sound trials in the D₁ session showed similar overall patterns between pre- and post-transition trials (with small differences in amplitudes). On the other hand, BF activity in error sound trials was significantly reduced compared to post-transition correct trials. This inhibition occurred in the epoch between fixation port exit and outcome, dipping below the baseline firing rates of BF bursting neurons. BF activity was further inhibited at the outcome epoch in error trials when no reward was delivered. Notice that the BF phasic bursting response to the onset of the sound did not differ between correct and error trials.

These results overall support that the behavioral performance and BF neural responses were relatively stable in sound trials after the abrupt transition in the D₁ session. In the minority of trials when animals made the incorrect choice to lick at the right reward port, BF activity was significantly inhibited prior to receiving the outcome.

Figure S3: Behavioral and BF response patterns in sound trials during the D₁ session

A, Behavioral response patterns in sound trials, catch trials and light trials during the D₁ session, aligned at the behavioral transition point identified in Figure 4. All 7 animals were included in this analysis. Since the three trial types were presented at equal probabilities, the number of trials in each trial type was roughly the same. **B**, Single trial BF responses in sound trials in the example D₁ session from Figure 3E2, aligned at stimulus onset and

outcome. BF activities at the two epochs were calculated at [0.05, 0.2]s after sound onset and [0.05, 0.35]s after 3rd lick. Sound trials with incorrect rightward licks were highlighted. **C**, Average BF activities aligned at the three behavioral events, plotted separately for pre-transition correct trials, post-transition correct trials, and post-transition error trials. Horizontal lines indicate significant differences in population BF activities ($p < 0.01$ for 3 consecutive steps using paired t-test for each 100 ms sliding window and 10ms step). Notice that BF activities in error sound trials were significantly inhibited prior to receiving the outcome.

-What does responding to auditory cued trials look like, and is it disrupted at all, behaviorally or neurophysiologically, upon introducing the light cue?

Please see our response above (**Figure S3**).

-Shouldnt the left hand side of E2 show what the response profile was to the auditory-go right situation prior to transitioning? Rather than showing no-lick trials?

Thank you for the comment. Our goal in Figure 3E2 was to highlight the changes in BF activity associated with the abrupt behavioral transition in the three trial types (light, catch and no-fix licks) that showed rightward licking. That was the reason why we showed the light:no-lick and catch:no-lick trials for comparison. The behavioral and neuronal response patterns of the sound trial remained relatively stable as discussed above (**Figure S3**).

-What do the non-burst neurons do, and why are they not being shown in contrast to the burst population?

Thank you for the question about other BF neurons. We showed the firing rate modulations toward the major behavioral events of all recorded BF neurons in Figure S2. BF neurons that were not classified as bursting neurons (30%, 440/1453) are a heterogeneous group, in terms of baseline firing rates, firing pattern statistics, and task-related modulations. Given their heterogeneity, we do not have enough sampling in the current study for any subtype that would allow us to draw strong conclusions. An extensive characterization of other BF neurons in this type of task has been reported in an earlier publication (Avila & Lin, Front in Beh Neurosci, 2014), which is also reproduced in **Reviewer-only Figure R3** for comparison with results from the current study. How the heterogeneous response profiles of other BF neurons correspond to the many cell types in the BF will be an important topic in the future.

Figure R3: Compare BF activities in catch trials from two studies

A, Figure 2B from Avila & Lin (Front in Beh Neurosci, 2014) shows the response of three types of BF neurons to different behavioral events in a similar task with only one reward port. Type I neurons correspond to BF bursting neurons, while the two other types of BF neurons show activity modulations related to fixation. The three middle columns show BF activities aligned at events in catch trials: [fixation entry, 'catch' stimulus, fixation exit]. The fifth column compares BF activity at stimulus onset between sound trials and catch trials (black). **B**, BF activity in catch:no-lick trials in the D₁ session (from Figure 5A), aligned at the same events as in panel A. The activity patterns of BF bursting neurons in catch no-lick trials are highly similar in these two studies (gray boxes).

Behavior

--Since there are four different foreperiods in the fixation port prior to a stimulus being delivered (or not in the case of catch), what was the activity of BF neurons as those periods expired without a stimulus, if any?

Thank you for the question about BF activities in catch trials. BF activity in catch trials decreased during the foreperiod while animals were waiting in the fixation port. This can be best visualized in Figure 5A (replotted here in **Figure R3B**). On average, BF activity dipped below baseline firing rates right around the timing of the would-be-stimulus. In catch no-lick trials, BF activity continued to be inhibited until rats exited the fixation port, and slowly returned back to baseline firing rates.

-When in a catch trial, did rats wait typically to a time past the longest foreperiod before answering exiting and entering a side port?

We conducted a new analysis to address this issue (**Reviewer-only Figure R4**). The median fixation duration in catch trials remained stable across sessions, averaging 1.30s in catch no-lick trials and 1.17s in catch lick trials, both of which were substantially longer than the longest foreperiod at 0.8s. The median fixation durations in catch no-lick trials were not different from those in light no-lick trials. The pattern was different in light lick trials: the fixation duration in light lick trials was the same as catch lick trials in the D₁ session (when rats had not learned about the light stimulus), and decreased significantly in subsequent

sessions as rats learned about the light stimulus. These patterns indicate that rats indeed waited past the longest foreperiod before exiting the fixation port in catch trials and this behavior was stable throughout the learning process.

Figure R4: Fixation duration in catch and light trials

Median fixation duration in catch and light trials were plotted separately for no-lick trials (A) and rightward licking trials (B) across sessions. Each circle indicates one session, and sessions from the same animal were connected by a thin line. Thick lines indicate group averages. Sessions were aligned at the D_1 session of each animal. The longest foreperiod (0.8s) is shown as the horizontal dotted line. Significant differences between catch and light trials (paired t-test) are shown above. Trial types with fewer than 20 trials in a session were excluded from this analysis.

-Isn't there a fourth type of right lick? Where it is an auditory cued trial, but the animal gets it incorrect by licking right? In those cases, what happens? Or are there too few as these trials?

The error rate in sound trials (i.e. incorrect rightward licking) was $6.4 \pm 3.3\%$ across sessions (mean \pm std, $N=7$ rats) and BF activity was significantly lower in the error trials. Please see our discussion of sound trials above (Figure S3).

The emergence and subsequent elimination of non-rewarded behaviors were mirrored by changes in BF activity

Fig 4a

-Figure 4 should include a column aligned to side port (right port) entry. My question is whether BF activation may be more precisely related to side port entry than to fixation port exit, and indicate the state of being in the side port.

Please see our responses regarding the alignment to side (reward) port entry (Figure S4, Figures R1-R2).

BF neurons did not respond to the new light stimulus during initial learning

Fig5a

-Here too Figure 5 should include a column aligned to side port (right port) entry. My sense is that BF activation may be more related to side port entry than to fixation port exit, and indicate the state of being in the side port.

Please see our responses regarding the alignment to side (reward) port entry (**Figure S4, Figures R1-R2**).

BF responses to light onset emerged later when the light stimulus was used to guide reward-seeking behavior

-6c left: I'm surprised from the data shown in 6c that early light v early catch does not show a significant difference. Can the authors confirm? It appears as if it should be different, with the duration of the response extending in the later light trials over that of the early light trials. In any case, the absence of light responses in D1 indicate that light-evoked responses emerge as light acquires predictive power.

Thank you for the important question. After considering your question as well as comments from Reviewer #1, we have modified Figure 6C and provided additional analyses to better understand how BF responses to the new light emerged in the D_2 session. We first compared BF responses in light lick and catch lick trials between three parts of the learning curve: (1) late trials in the D_{2-1} session; (2) early trials in the D_2 session; (3) late trials in the D_2 session. At the end of the D_{2-1} session, BF neurons did not show response to the new light (paired t-test between light lick and catch lick trials, $p=0.51$, $N=6$). In early D_2 trials, BF phasic responses to the new light were clearly visible in 4 animals. Comparison between late trials in the D_{2-1} session and the early D_2 trials showed that there was a trend toward increasing BF responses (Figure 6C1, significant at $p<0.05$ level; Figure 6C2, paired t-test, $p=0.065$, $N=6$). These observations suggest that the BF phasic response to the new light could develop offline in between sessions. In addition, during the D_2 session, BF responses to the new light increased in all animals between early and late trials (Figure 6C2, paired t-test, $p=0.003$, $N=7$). This increase took place mostly during the sustained phase of the BF response that was better aligned with fixation port exit than with stimulus onset (Figure 6D). Together, these new results extend our observation in Figure 6B that BF responses to the new light developed in the D_2 session by suggesting that BF responses to the new light partly developed offline (between D_{2-1} and D_2 session), and partly strengthened during the D_2 session.

Figure 6C. The emergence of BF responses to the new light in the D₂ session.

C1, BF activities in light lick and catch lick trials at three points of the learning process: the last 20 trials (late) in the D₂-1 session; first 20 trials (early) in the D₂ session; late trials in the D₂ session. Top row depicts BF activities in light lick trials from individual animals, while the bottom row depicts population BF activities (mean \pm s.e.m.) in light lick and catch lick trials (N=6-7). Significant differences in BF activities between the two trial types were indicated by horizontal lines (red: $p < 0.01$, 3 consecutive bins; pink: $p < 0.05$, 3 consecutive bins). Yellow shaded intervals indicate the time windows for calculating BF responses to the new light in C2. **C2**, Average BF responses to the new light, defined as the difference between the two trial types, at the three points of the learning process (paired t-test, N=6-7). **D**, Comparison of BF activities between early and late light lick trials in the D₂ session, aligned at stimulus onset and fixation port exit. Top row shows the activity difference in individual animals, and the bottom row shows population BF activities in the two trial types (mean \pm s.e.m.) (N=7). Significant differences in BF activities ($p < 0.01$, 3 consecutive bins) were indicated by horizontal lines. The activity difference was stronger and better aligned at fixation port exit.

-The suppression of the reward outcome-elicited response, and the augmentation to the reward predictive light stimuli are consonant with RPE, but how is elevated BF activity over the 'evaluation' period a sign of reward prediction error? Does the reward come at a random delay once in the side port? If so, then yes, could be a sign of RPE over that interval. Hmm- it doesn't come at a random delay, but rather on the occurrence of the 3rd lick. What is the distribution of the 3rd lick times across the session? This may then form a probability density function of reward delays from fix-out (or maybe better side port in) experienced by the animal, giving rise under a RPE framework for why there is a BF response across the 'evaluation' period.

Thank you for the question. To better understand the timing of the 3rd lick, we calculated the median interval duration between [fix-out, 3rd lick], [reward port entry, 3rd lick] and [1st lick, 3rd lick] in light lick and catch lick trials (**Reviewer-only Figure R5**). The [fix-out, 3rd lick] interval showed significant modulation during the new learning phase in light lick trials. In contrast, the [reward port entry, 3rd lick] and [1st lick, 3rd lick] intervals were stable across new learning sessions. The stereotypical behavioral pattern after entering the reward port therefore could allow animals to form strong expectations of when the reward would be delivered.

Figure R5: Interval durations before licking at the right reward port in light lick and catch lick trials.

Median interval durations between [fix-out, 3rd lick] (**A**), [reward port entry, 3rd lick] (**B**) and [1st lick, 3rd lick] (**C**) in light lick and catch lick trials. Each circle indicates one session, and sessions from the same animal were connected by a thin line. Thick lines indicate group averages. Sessions were aligned at the D₁ session of each animal. Significant differences between catch lick and light lick trials (paired t-test) are indicated. Trial types with fewer than 20 trials in a session were excluded from this analysis.

Stronger BF responses to the new light reflected better learning and faster decisions

“plateaued” change to “plateau”

Thank you! Edited as suggested.

Citations

Additional papers assisting in the case that BF conveys reinforcement prediction error

(Hegedüs et al. 2023; Chubykin et al. 2013; Liu et al. 2015)

These references have been added to the discussion, where we discuss the role of other neuromodulatory systems in the encoding of reward prediction error.

Chubykin, Alexander A., Emma B. Roach, Mark F. Bear, and Marshall G. Hussain Shuler. 2013. “A Cholinergic Mechanism for Reward Timing within Primary Visual Cortex.” *Neuron* 77 (4): 723–35.

Hegedüs, Panna, Katalin Sviatkó, Bálint Király, Sergio Martínez-Bellver, and Balázs Hangya. 2023. "Cholinergic Activity Reflects Reward Expectations and Predicts Behavioral Responses." *IScience* 26 (1): 105814.

Liu, C. H., J. E. Coleman, H. Davoudi, K. Zhang, and M. G. Hussain Shuler. 2015. "Selective Activation of a Putative Reinforcement Signal Conditions Cued Interval Timing in Primary Visual Cortex." *Current Biology: CB* 2015/05/26 (12): 1551–61.

REVIEWERS' COMMENTS

Reviewer #1 (Remarks to the Author):

The Authors did an excellent job in addressing my points. Regarding the novel analyses, I find the possibility that the BF phasic response to the new light could develop offline in between sessions particularly exciting. I have no further concerns about this important study.

Balazs Hangya

Reviewer #2 (Remarks to the Author):

The authors have thoroughly and clearly addressed each comment of the review, and also provided additional analyses that further deepen the findings of this important work.

After careful re-reading of their response to reviewers' comments and the revised manuscript, I have no further concerns.

REVIEWERS' COMMENTS

Reviewer #1 (Remarks to the Author):

The Authors did an excellent job in addressing my points. Regarding the novel analyses, I find the possibility that the BF phasic response to the new light could develop offline in between sessions particularly exciting. I have no further concerns about this important study.

Balazs Hangya

Reviewer #2 (Remarks to the Author):

The authors have thoroughly and clearly addressed each comment of the review, and also provided additional analyses that further deepen the findings of this important work.

After careful re-reading of their response to reviewers' comments and the revised manuscript, I have no further concerns.

We thank both reviewers for the constructive comments during the review process. No further concerns need to be addressed at this stage.